# Kick-starting the zygotic genome: licensors, specifiers, and beyond

Zhuoning Zou[1,6], Qiuyan Wang[1,6], Xi Wu[1,2], Richard M Schultz [3,4] & Wei Xie [1,5✉]

## Abstract

**Zygotic genome activation (ZGA), the first transcription event following fertilization, kickstarts the embryonic program that takes over the control of early development from the maternal products. How ZGA occurs, especially in mammals, is poorly understood due to the limited amount of research materials. With the rapid development of single-cell and low-input technologies, remarkable progress made in the past decade has unveiled dramatic transitions of the epigenomes, transcriptomes, proteomes, and metabolomes associated with ZGA. Moreover, functional investigations are yielding insights into the key regulators of ZGA, among which two major classes of players are emerging: licensors and specifiers. Licensors would control the permission of transcription and its timing during ZGA. Accumulating evidence suggests that such licensors of ZGA include regulators of the transcription apparatus and nuclear gatekeepers. Specifiers would instruct the activation of specific genes during ZGA. These specifiers include key transcription factors present at this stage, often facilitated by epigenetic regulators. Based on data primarily from mammals but also results from other species, we discuss in this review how recent research sheds light on the molecular regulation of ZGA and its executors, including the licensors and specifiers.**

**Keywords** ZGA; Early Embryo; Licensor; Specifier; Transcription Factor
**Subject Categories** Chromatin, Transcription & Genomics; Development

## Introduction

In all species examined so far, following fertilization, there is a period when transcription is absent prior to the onset of ZGA at a defined developmental stage. ZGA usually entails two waves of transcription, minor ZGA, when a small set of genes are activated, and major ZGA, when thousands of genes are transcribed (Jukam et al, 2017). In mouse, minor ZGA and major ZGA occur during middle-1-cell and late 2-cell stages, respectively (Jukam et al, 2017; Lee et al, 2014). Both the transcription products and transcription per se likely play critical roles in embryonic development (Abe et al,

2018; Golbus et al, 1973; Liu et al, 2020). During ZGA, epigenetic landscapes undergo drastic reprogramming to accommodate these first transcriptional events. Key transcription factors (TFs) are usually considered as the driving force of ZGA to activate specific genes. However, loss of master TFs only causes partial failure of ZGA programs, and premature expression of TFs usually is insufficient to cause advanced ZGA (Larson et al, 2022). Therefore, in addition to key TFs serving as crucial "specifiers" in regulating ZGA genes, other factors may act as ZGA "licensors" that control permission of transcription by creating a permissive environment for transcription without exhibiting strong gene selectivity (Fig. 1). In this review, we first review recent progress in our understanding of the molecular nature underlying ZGA. We then discuss how "licensors", including regulators of the transcription apparatus and nuclear gatekeepers, generate competency for ZGA and possibly control its timing. We finally discuss "specifiers", key ZGA TFs, and their impact on embryonic development.

## Epigenetic reprogramming during ZGA

Dramatic reprogramming of epigenetic landscapes occurs during ZGA that involves resetting of epigenetic marks, chromatin accessibility, and chromatin organization (Fig. 2) (Du et al, 2022; Vallot and Tachibana, 2020). Some of these reprogramming events ensure the onset of ZGA or the fidelity of ZGA, whereas others are likely consequences of ZGA. Here we first review the dynamics and possible roles of DNA methylation, histone modifications and histone variants, chromatin remodelers, and 3D chromatin organization in ZGA.

### DNA methylation

In both mouse and human, the paternal genome undergoes global DNA demethylation rapidly after fertilization, while the maternal genome gradually lose DNA methylation (Fig. 2), leaving 20–40% of CpG sites with gamete-inherited methylation in blastocysts (Iurlaro et al, 2017). Exceptions include imprinting control regions and certain repeats, which retain high levels of DNA methylation (Greenberg and Bourc'his, 2019; Iurlaro et al, 2017). The transiently inherited oocyte methylation in 1-cell embryos can shape the landscape of initial RNA polymerase II (POLR2/Pol II)

[1]Center for Stem Cell Biology and Regenerative Medicine, MOE Key Laboratory of Bioinformatics, School of Life Sciences, Tsinghua University, 100084 Beijing, China. [2]Peking University-Tsinghua University-National Institute of Biological Sciences (PTN) Joint Graduate Program, Academy for Advanced Interdisciplinary Studies, Peking University, 100871 Beijing, China. [3]Department of Biology, University of Pennsylvania, Philadelphia, PA, USA. [4]Department of Microbiology and Molecular Genetics, College of Biological Sciences, University of California, Davis, Davis, CA, USA. [5]Tsinghua-Peking Center for Life Sciences, Beijing, China. [6]These authors contributed equally: Zhuoning Zou, Qiuyan Wang. ✉E-mail: xiewei121@tsinghua.edu.cn

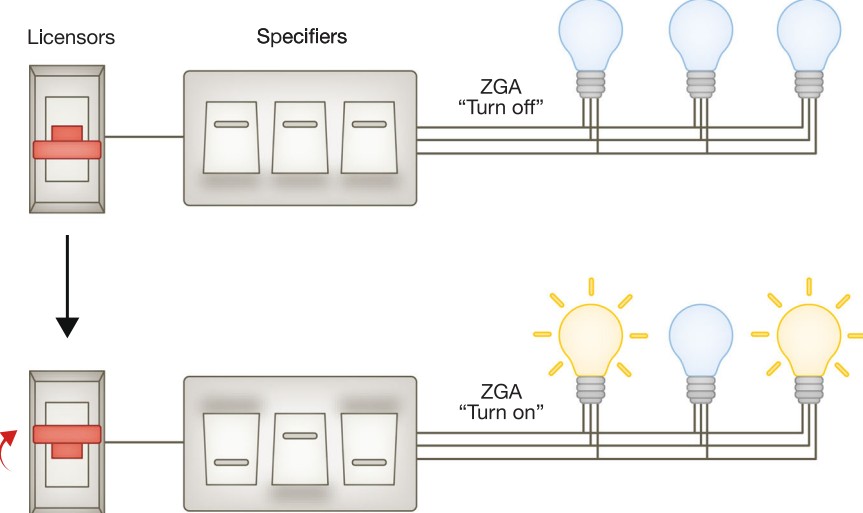

**Figure 1. ZGA "licensor" and "specifier" model.**

The electric lever, switches, and lamps serve as a model for illustrating the process of ZGA. In this analogy, ZGA "licensors" act as the electric lever, governing the flow of electricity (permission of transcription). ZGA "specifiers" function as distinct switches, regulating the activation of specific lamps (genes).

binding (Liu et al, 2020). This is not expected to have a major impact on ZGA though, as *Dnmt3a* or *Dnmt3l* maternal knockout (mKO) embryos, which lose all DNA methylation in oocytes, can develop beyond the pre-implantation period and only exhibit lethality after implantation, presumably due to imprinting defects (Dahlet et al, 2020; Okano et al, 1999) (Bourc'his et al, 2001; Hata et al, 2002). On the other hand, aberrant acquisition of DNA methylation in oocytes due to abnormal nuclear retention of DNMT1 and its cofactor UHRF1 upon loss of *Stella* impeded activation of some zygotic genes after fertilization (Li et al, 2018b). Aberrant nuclear retention of DNMT1/UHRF1 was also found in mouse zygotes with depleted maternal NLRP14, an NLRP (Nucleotide-binding oligomerization domain, Leucine-rich Repeat, and Pyrin domain containing) family protein, coinciding with impaired ZGA (Yan et al, 2023). Unlike *Stella* mutants, *Nlrp14* mutants exhibited defective passive demethylation but not ectopic DNA methylation. NLRP14 also has DNA methylation-independent functions as oocytes lacking NLRP14 showed impaired mitochondria dynamics and the resulting embryos displayed disrupted $Ca^{2+}$ oscillations upon fertilization (Meng et al, 2023). Interestingly, not all DNA methylation dynamics are unidirectional (demethylation) in the early embryo, as a few paternal regions are also subjected to DNMT3A-dependent de novo methylation starting from the 2-cell stage (Richard Albert et al, 2020). Loss of such de novo methylation leads to premature activation of these genes from the paternal genome at the 4-cell stage (Richard Albert et al, 2020). Taken together, these data indicate that proper DNA methylation reprogramming promotes the fidelity of gene expression during ZGA and afterwards (Fig. 2).

## Histone modifications

### *Histone acetylation*
Histone acetylation marks active promoters and enhancers and can promote the opening of transcribed chromatin in part by

neutralizing the positive charge of histone tails (Bannister and Kouzarides, 2011). H3K27ac, one widely studied histone acetylation site, is found at promoters before ZGA in zebrafish (Zhang et al, 2018a), *Drosophila* (Li et al, 2014), and mouse (Dahl et al, 2016; Wang et al, 2022b; Liu et al, 2024) embryos. In mouse, major ZGA genes are primed with histone acetylation in zygotes and early 2-cell embryos (Wang et al, 2022b). H3K27ac, which is hypoacetylated in MII eggs, appears as a non-canonical broad pattern and correlates with H3K4me3 and chromatin accessibility in germinal vesicle oocytes and zygotes before its distribution is reprogrammed during the 2-cell stage (Wang et al, 2022b) (Fig. 2). After fertilization, H3K27ac is de novo established on the paternal genome earlier than that on the maternal genome, also manifesting as larger domains (Fig. 2) (Wang et al, 2022b). In human, H3K27ac exhibits a broad distribution pattern in 2-cell and 4-cell embryos prior to ZGA, correlates with H3K4me3 during the 4-cell stage, and switches to a canonical sharp pattern during the 8-cell stage (Wu et al, 2023; Xia et al, 2019). In zebrafish, chemical inhibition of P300, the writer of histone acetyl-lysine, or BRD4, the reader of histone acetyl-lysine, or knockdown of histone acetyltransferases Ep300b, Crebbpa, and Crebbpb, causes a reduction of zygotic transcription and arrested gastrulation (Chan et al, 2019; Zhang et al, 2018a). Artificial recruitment of p300 activates ZGA genes and bypasses the need for TFs (Miao et al, 2022). Similarly, chemical inhibition of CBP/P300 leads to defective ZGA, loss of chromatin accessibility, and 2-cell arrest in mouse (Wang et al, 2022b). Of note, loss of histone acetylation at H3K27 alone was insufficient to affect gene activation in mouse embryonic stem cells (mESCs), presumably due to the presence of other histone acetylations (such as H3K9ac, H3K18ac, H4K5ac, H4K8ac, etc.) (Sankar et al, 2022; Zhang et al, 2020b). In fact, histone acetylation occurs at a large number of lysine sites and is known to exert functions often in a partially redundant manner (Durrin et al, 1991; Zhang et al, 1998). In sum, these findings indicate that histone acetylation is required and, in some contexts, sufficient for the transcriptional activation during ZGA.

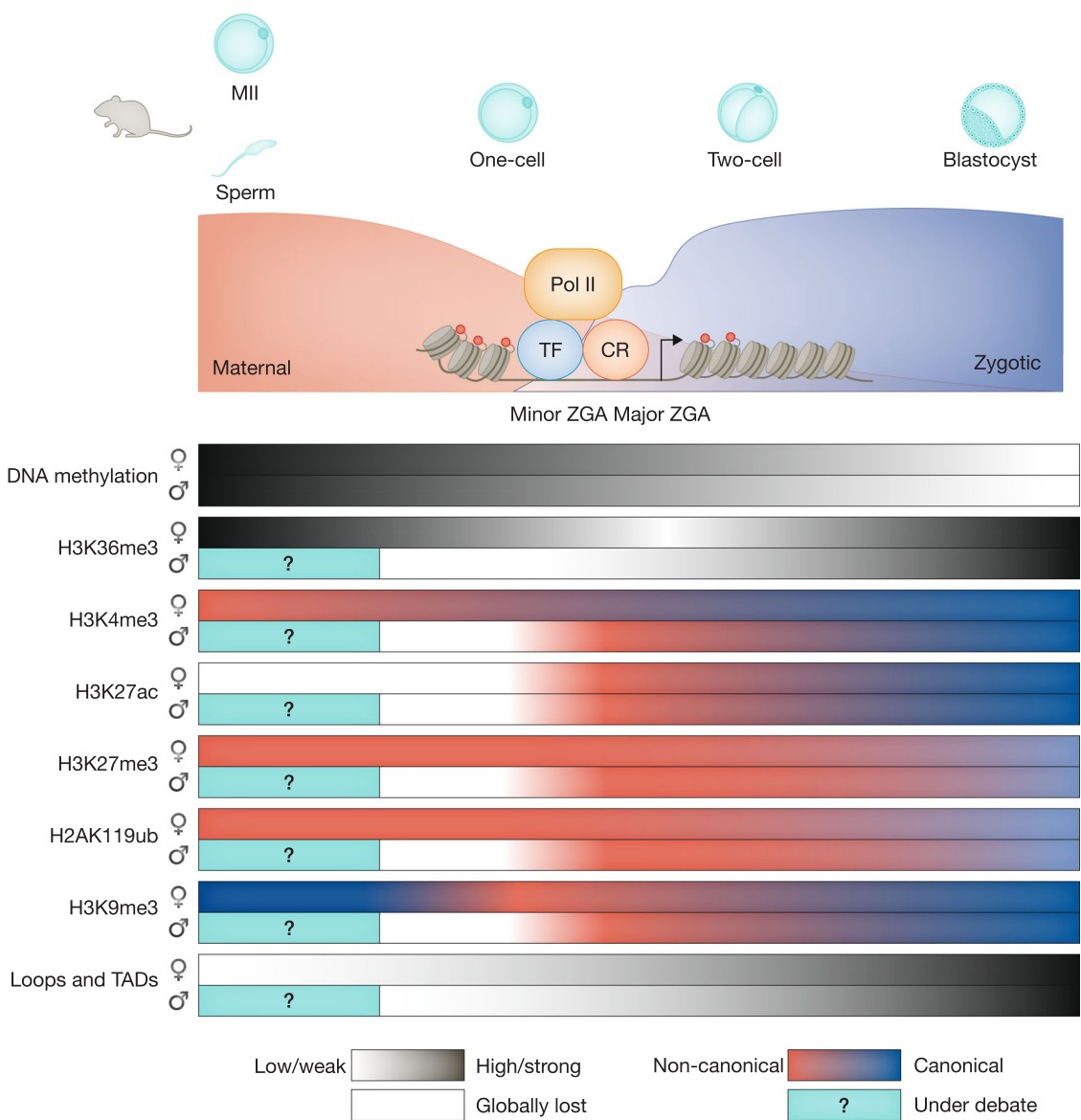

**Figure 2. Epigenetic reprogramming around mouse ZGA.**

In mouse, ZGA is executed by RNA Pol II and facilitated by transcription factors (TFs) and chromatin regulators (CRs), amid dramatic epigenetic reprogramming. After fertilization, genome-wide loss of DNA methylation starts from the zygote stage and reaches the lowest level in blastocysts. On the paternal genome, H3K36me3, H3K4me3, H3K27me3, H3K27ac, and H2AK119ub are quickly reset after fertilization based on the globally distinct patterns between sperm and late 1-cell embryo, consistent with the protamine-histone exchange that occurs shortly following fertilization. From the maternal genome, H3K36me3 is briefly inherited by 1-cell embryos and is then reset after ZGA to mark newly transcribed genes. H3K27ac is absent from chromatin of MII eggs and is re-established around pronuclear stage 3-4 (PN3-4) after fertilization. H3K4me3 is briefly inherited from oocytes to early 2-cell embryos. H3K27ac and H3K4me3 then transit from non-canonical, broad domains to canonical sharp peaks during ZGA. H3K27me3 and H2AK119ub exhibit broad, non-canonical patterns in oocytes and preimplantation embryos, with H3K27me3 domains away from promoters being inherited to blastocysts and H2AK119ub1 exhibiting more dynamic changes. Both marks switch to canonical patterns with sharp peaks at promoters after implantation (not shown here). The global patterns of H3K9me3 on both alleles show distinct distributions in 1-cell embryos compared to those in gametes, indicating epigenetic resetting on both genomes. H3K9me3 at early stages is considered non-repressive and is thus marked as non-canonical. 3D chromatin structures including loops and topologically associating domains (TADs) are substantially reduced in MII eggs, 1-cell and 2-cell embryos but are gradually re-established after ZGA. The presence of loops, TADs, as well as the patterning of histone modifications in mouse sperm are currently under debate (Yin et al, 2023).

Interestingly, overexpression of an HDAC1/2 dominant-negative mutant or inhibition of HDAC activity also led to ZGA defects and aberrant expression of some developmental genes (Dang et al, 2022a; Wang et al, 2022b; Wu et al, 2023). In addition, nicotinamide adenine dinucleotide (NAD +), as the cofactor of histone deacetylation, is critical for the precise and timely control of minor ZGA in mouse (Li et al, 2022). These data suggest that histone deacetylation also contributes to mouse ZGA and preimplantation development (Fig. 2) (Wang et al, 2022b), although indirect effects from inhibitors and dominant mutants cannot be excluded. These results are also consistent with the previous proposal that chromatin repression is established in late

2-cell mouse embryos based on the finding that enhancers are necessary for plasmid-borne reporter activation in late 2-cell embryos but not in 1-cell embryos (Majumder et al, 1993).

### H3K4me3

H3K4me3 is a classic transcription-permissive histone mark preferentially found at promoters (Rao and Dou, 2015). In *Xenopus* (Akkers et al, 2009; Hontelez et al, 2015), zebrafish (Lindeman et al, 2011; Vastenhouw et al, 2010; Zhang et al, 2018a), and mouse (Dahl et al, 2016; Liu et al, 2016; Zhang et al, 2016), H3K4me3 is already present prior to ZGA (Fig. 2), whereas it is absent in *Drosophila* embryos before ZGA, suggesting the ZGA in *Drosophila* does not require H3K4me3 (Chen et al, 2013b). H3K4me3 appears as a non-canonical pattern in oocytes of mouse, rat, bovine, and porcine, in the form of broad domains at promoters and many distal regions that could span more than 10 kb (Dahl et al, 2016; Hanna et al, 2018; Lu et al, 2021b; Zhang et al, 2016). The function of non-canonical H3K4me3 domains remains largely unclear. One hint is that it may play a role in genome silencing in SN (surrounded nucleolus) fully-grown mouse oocytes that are normally transcriptionally quiescent, because removal of H3K4me3 by overexpressing histone lysine demethylase KDM5B or loss of KMT2B, the methyltransferase responsible for a large fraction of H3K4me3 broad domains in oocytes, results in transcription reactivation, based on BrU staining (Andreu-Vieyra et al, 2010; Zhang et al, 2016). H3K4me3 appears as a canonical pattern in mouse sperm, although this is still debated (Yin et al, 2023). After fertilization, H3K4me3 on the paternal genome is quickly reduced and is re-established in a non-canonical pattern at the late one-cell stage in mouse (Zhang et al, 2016). Such resetting is possibly related to global nucleosome disassociation during protamine-histone exchanges, given similar resetting has been observed for multiple histone marks (Fig. 2) (Du et al, 2022), although direct evidence is still lacking. At the two-cell stage, non-canonical H3K4me3 from both alleles switches to a canonical pattern (Fig. 2), being restricted to CpG-rich regions preferentially at promoters (Dahl et al, 2016; Zhang et al, 2016). This transition requires zygotic transcription, in part because KDM5A/B, two histone lysine demethylases involved in this pattern switch, are expressed during ZGA (Dahl et al, 2016). Knocking down *Kdm5a/5b* leads to dysregulated ZGA and development fails at the blastocyst stage (Dahl et al, 2016), suggesting that H3K4me3 redistribution is important for mouse embryonic development. A similar requirement for KDM5 for early embryonic development was also reported in bovine and porcine (Bu et al, 2022; Dang et al, 2022b). Intriguingly, removal of H3K4me3 by overexpression of KDM5B is accompanied by loss of nuclear lamina-associated domains of the paternal genome in mouse embryos (Borsos et al, 2019), suggesting that paternal H3K4me3 is somehow involved in establishing the initial higher-order chromatin architecture. Interestingly, non-canonical H3K4me3 does not exist in human oocytes and embryos, except that broad H3K4me3 domains briefly appear in CpG-rich, accessible regions at the 4-cell stage prior to major ZGA (Wu et al, 2023; Xia et al, 2019). Therefore, the function of non-canonical H3K4me3 and why it is missing in human await further studies.

### H3K27me3 and H2AK119ub

H3K27me3 and H2AK119ub are repressive histone marks deposited by Polycomb repressive complex 2 (PRC2) and Polycomb repressive complex 1 (PRC1), respectively. Both marks play a role in preventing erroneous gene activation, particularly for key developmental regulator genes (Blythe and Wieschaus, 2015; Margueron and Reinberg, 2011). In mouse, both H3K27me3 and H2AK119ub exhibit a non-canonical broad pattern in oocytes (Fig. 2) (Chen et al, 2021a; Zheng et al, 2016; Rishi and McManus, 1987; Du et al, 2022). After fertilization, H3K27me3 on the maternal genome is largely erased from promoters of developmental genes, while H3K27me3 on the paternal genome is first globally depleted and then gradually established in a non-canonical pattern as megabase-long broad domains in mouse zygotes (Zheng et al, 2016). As a result, H3K4me3 and H3K27me3 both form extraordinarily large domains but occupy gene-dense and gene-desert regions in the paternal genome, respectively (Zheng et al, 2016). These differences in distribution may reflect basal activities of activating and repressive histone modifiers in a naïve chromatin setting after the protamine-histone exchange. Unlike H3K27me3, H2AK119ub is present at promoters of developmental genes throughout mouse development (Chen et al, 2021a; Mei et al, 2021; Zhu et al, 2021). Depletion of H2AK119ub from the 1-cell stage causes 4-cell arrest and derepression of some developmental genes during mouse ZGA (Chen et al, 2021a), indicating that H2AK119ub helps to prevent precocious gene expression in the absence of H3K27me3. Moreover, depleting USP16, the major deubiquitinase for H2AK119ub in mouse oocytes, leads to excessive H2AK119ub and impaired ZGA (Rong et al, 2022). On the other hand, distal maternal H3K27me3 domains (away from promoters) persist to the blastocyst stage and play a role in DNA methylation-independent non-canonical imprinting for a small subset of genes including *Xist* in mouse (Inoue et al, 2017; Zheng et al, 2016; Deaton and Bird, 2011). In contrast, H2AK119ub is more dynamic than H3K27me3 throughout preimplantation development, with maternal and paternal alleles reaching similar patterns at the 2-cell stage (Chen et al, 2021a). After implantation, H3K27me3 and H2AK119ub both adopt similar distributions and canonical patterns, co-occupying developmental gene promoters (Fig. 2) (Chen et al, 2021a; Liu et al, 2016; Mei et al, 2021; Zheng et al, 2016; Zhu et al, 2021). Therefore, H3K27me3 appears to be briefly lost at key Polycomb target gene promoters in preimplantation embryos, leading to resetting of the epigenetic memory from gametes to embryos. H2AK119ub may provide necessary repression to ensure that these genes do not undergo precocious activation in this developmental window, including ZGA. This proposal is consistent with the notion that H3K27me3 is more stable than H2AK119ub and is thus more suitable to serve as long-term epigenetic memory (Moussa et al, 2019).

### H3K9me3

H3K9me3 is a repressive mark that is primarily deposited in constitutive heterochromatin (Allis and Jenuwein, 2016) and is a major epigenetic barrier for iPSC reprogramming (Chen et al, 2013a; Onder et al, 2012). In mouse, ChIP-seq analyses revealed global resetting of H3K9me3 on both maternal and paternal alleles in 1-cell embryos (Wang et al, 2018) (Fig. 2). SUV39H1, an H3K9me3 histone methyltransferase, was undetectable by immunostaining until the 8-cell stage. SUV39H2 is maternally derived from oocytes and catalyzes de novo H3K9me3 on the paternal genome (Burton et al, 2020). Interestingly, this H3K9me3 is non-repressive and is compatible with gene expression because only a few genes are upregulated or downregulated in late 2-cell embryos after *Suv39h2* knockdown in zygotes (Burton et al, 2020). In mouse embryos, erasure of H3K9me3 after fertilization correlated with

expression of transiently activated 2-cell genes, e.g., *Zscan4*, and retrotransposons, e.g., MERVL. Subsequent establishment of H3K9me3 can silence long terminal repeats (LTRs) in the absence of DNA methylation (Wang et al, 2018). Improper reprogramming of H3K9me3 frequently occurs in somatic cell nuclear transfer (SCNT) embryos and results in insufficient gene activation during ZGA (Matoba et al, 2014), which could be rescued by over-expression of *Kdm4b/d/e* or knocking down *Suv39h1/2* in donor cells (Liu et al, 2018; Matoba et al, 2014; Xu et al, 2023). Depletion of KDM4A, an H3K9me3 demethylase, from oocytes, leads to ectopic accumulation of H3K9me3 and subsequently impaired activation of genes and transposable elements during ZGA (Sankar et al, 2020). On the other hand, overexpression of *Suv39h1* in mouse zygotes prematurely established constitutive heterochromatin and caused embryonic lethality after the morula stage (Burton et al, 2020). The H3K9me3 landscape was also characterized in human embryos (Xu et al, 2022; Yu et al, 2022a). Intriguingly, *Dux*, a ZGA gene, and multiple KRAB-ZNFs were identified as potential factors for establishing 8-cell- and blastocyst-specific H3K9me3. Knocking out *Dux* or knocking down *Zfp51* in mouse embryos attenuates stage-specific H3K9me3 deposition, indicating that ZGA helps to shape H3K9me3 reprogramming (Xu et al, 2022).

## Histone variants

Histone variants are incorporated into nucleosomes throughout the cell cycle in a replication-independent manner (Talbert and Henikoff, 2017). In zebrafish, "Placeholder" nucleosomes, which consist of histone variant H2A.Z and H3K4me1 but lack DNA methylation, are inherited from the gametes, and subsequently resolved into active or poised chromatin during ZGA (Murphy et al, 2018). Such "Placeholder" nucleosomes were proposed to facilitate DNA methylation reprogramming and precise execution of ZGA. In fly, H2A.Z, which is enriched at the promoters of 65% of ZGA genes, is deposited prior to Pol II loading (Ibarra-Morales et al, 2021). Maternal knockdown of H2A.Z histone chaperone and ATPase, Domino, leads to global reduction of H2A.Z at promoters, which further reduces Pol II occupancy, downregulates house-keeping genes, and compromises the establishment of 3D chromatin structures during ZGA (Ibarra-Morales et al, 2021). In mouse, H2A.Z is accumulated in gametes but is depleted in MII eggs, zygotes, and early 2-cell stage until it starts a prominent accumulation around ZGA in 2-cell embryos (Liu et al, 2022a). Consistent with previous reports and its multifaceted functions (Colino-Sanguino et al, 2022), H2A.Z is associated with both active and repressed genes in early embryos (Liu et al, 2022a). Knocking down H2A.Z by small interfering RNA (siRNA) affected lineage commitment and significantly reduced blastocyst formation, but did not have a strong effect on ZGA (Liu et al, 2022a), although it is unclear if residual H2A.Z is present and functions during ZGA. Regardless, these data indicate that once transcription starts, H2A.Z is an essential component of early development.

H3.3, which is the most conserved non-centromeric H3 variant (Shi et al, 2017; Strahl and Allis, 2000; Talbert and Henikoff, 2010), is present in promoters, gene bodies, and cis-regulatory elements of actively transcribed genes. H3.3 is also deposited on telomeres, pericentric heterochromatin, and silent retroviral elements, indicating that its role in transcription may depend on chromatin context (Shi et al, 2017). Genome-wide mapping showed that H3.3

exhibits a non-canonical pattern in mouse MII eggs and zygotes, where it distributes evenly across the genome but is depleted from promoters (Ishiuchi et al, 2021), reminiscent of the widespread distribution of H3.1/H3.2 in the genome (Filipescu et al, 2013). This pattern is consistent with the notion that H3.3 serves as the major H3 type in oocytes and 1-cell embryos given that canonical H3.1/H3.2 presumably cannot be incorporated into chromatin in the absence of DNA replication (Akiyama et al, 2011; Fulka et al, 2019; Nashun et al, 2015). This non-canonical pattern then quickly switches to a canonical pattern at the 2-cell stage, where H3.3 appears at regions marked by H3K4me3 and accessible chromatin. Interestingly, this transition occurs in a transcription-independent but replication-dependent manner (Ishiuchi et al, 2021) and is attributed to the replication-coupled H3.1/H3.2 deposition that presumably displaces H3.3. Either overexpression of a dominant-negative mutant of p150, a subunit of the H3.1/H3.2 chaperone CAF-1, in mouse embryos or knocking down p150 in mESCs led to increased incorporation of H3.3 likely through a "nucleosome gap-filling" mechanism for H3.3 (Ishiuchi et al, 2021; Ray-Gallet et al, 2011). Upon depletion of p150, "2-cell genes" such as *Zscan4* and *Dux* were activated, suggesting that the non-canonical H3.3 in 1-cell embryos represents a permissive chromatin environment that permits expression of minor ZGA genes (Ishiuchi et al, 2021; Ishiuchi et al, 2015). Interestingly, in *Drosophila*, an opposite trend appears to occur as canonical H3, supplied by the oocyte, constitutes most of the H3 in early cleaving embryos until around ZGA when H3.3 shows a gradual increase, accompanied by the reduction of canonical H3, likely caused by titration due to multiple rounds of DNA replication (Shindo and Amodeo, 2019). It is unclear why fly uses canonical H3 in oocytes. H3.3 mutant *Drosophila* can survive to adulthood and appear morphologically normal, though with reduced viability (Sakai et al, 2009). Without H3.3, the expression of canonical H3 is increased and canonical H3 is incorporated into transcribed regions in a replication-independent manner (Sakai et al, 2009). Neverthless, both male and female adult flies are sterile, suggesting H3.3 is specifically required for germline development (Sakai et al, 2009). Even in this case, defects in testis development were partially rescued by overexpression of canonical H3.2 (Sakai et al, 2009), suggesting that canonical histones can replace the function of H3.3 when expressed in large quantities (Klein and Knoepfler, 2023). One possibility is that flies produce canonical H3 in large quantities to compensate for H3.3 in oocytes to support the extremely rapid DNA replication cycles during embryonic development.

## Chromatin remodelers

Chromatin accessibility undergoes genome-wide reprogramming during ZGA and early development, as reviewed elsewhere (Du et al, 2022; Eckersley-Maslin et al, 2018). Chromatin remodelers are recruited by TFs or histone modification readers to alter nucleosome positions or conformations and chromatin accessibility (Wu and Vastenhouw, 2020). In late mouse zygotes, immuno-fluorescence analyses showed that SMARCA4 (BRG1) and SMARCA5 (SNF2H), components of the SWI/SNF chromatin remodeler complex, colocalized with transcription foci (Torres-Padilla and Zernicka-Goetz, 2006). Loss of maternal BRG1 led to a 2-4 cell arrest and reduced transcription of 30% of expressed genes in mouse embryos (Bultman et al, 2006). SMARCA5 depletion by

Trim-Away in mouse early zygotes also led to altered chromatin landscapes and defects in major ZGA, although these embryos formed blastocysts without obvious developmental defects (Kubinyecz et al, 2023). It remains to be determined is whether SMARCA5 recovered at later stages after transient Trim-Away depletion. Loss of BRG1-associated factor SMARCD2 also altered the chromatin accessibility of ~25% of protein-coding genes in zygotes and reduced blastocyst formation (Zhang et al, 2022). Chromatin remodelers may facilitate TF to overcome the "barrier" of the chromatin state, as genes with less BRG1-dependency tended to be more accessible and activated more rapidly compared to those with higher BRG1-dependency in mESCs (King and Klose, 2017). Chromatin remodelers can also help to maintain TF binding because sustained OCT4/SOX2/NANOG binding continually requires BRG1 in mESCs (King and Klose, 2017). Depletion of chromatin remodelers typically did not affect global histone acetylation levels by immunostaining (Bultman et al, 2006; Torres-Padilla and Zernicka-Goetz, 2006), which indicates they might act downstream or in parallel with histone acetyltransferases.

## 3D chromatin organization

A hallmark of chromatin organization in early embryos is the extensively relaxed chromatin organization after fertilization, identified by HiC, which is highly conserved among different species (Vallot and Tachibana, 2020; Hug and Vaquerizas, 2018; Zhang and Xie, 2022), including fly (Hug et al, 2017; Ogiyama et al, 2018), fish (Kaaij et al, 2018; Nakamura et al, 2021; Wike et al, 2021), *Xenopus* (Niu et al, 2021), pig (Li et al, 2020), mouse (Collombet et al, 2020; Du et al, 2017; Flyamer et al, 2017; Ke et al, 2017), and human (Chen et al, 2019). In mouse, loops, topologically associating domains (TADs), and compartments are all substantially weakened in zygotes and two-cell embryos compared with later stages (Collombet et al, 2020; Du et al, 2017; Zhang et al, 2018b; Zheng and Xie, 2019), although residual TADs and loops were still detectable in zygotes and depend on cohesin and CTCF (Flyamer et al, 2017; Gassler et al, 2017) (Fig. 2). Why TADs and compartments become highly relaxed remains unclear, but it may provide a more permissive environment for ZGA and also reset the epigenetic memories from gametes encoded in chromatin organization. In human embryos, CTCF, which is required for TAD formation (Chen et al, 2019), is expressed upon ZGA, correlating with the emergence of TADs. This mechanism, however, does not appear to apply to the gradual reestablishment of TADs in mouse (Du et al, 2017; Hug et al, 2017; Ke et al, 2017), as CTCF is expressed throughout mouse early development, and binds and insulates chromatin in mouse zygotes (Dequeker et al, 2022; Wang et al, 2024).

The functions of CTCF and cohesin in ZGA and transcription regulation in early embryos remain a topic under investigation. Depletion of CTCF using a zona pellucida 3 (*Zp3*) promoter-driven double-stranded RNA (dsRNA) leads to defects in meiosis, development delay at 2 ~ 4 cell stage and disruption of ZGA in mice (Wan et al, 2008). However, a more recent study reported that maternal-zygotic *Ctcf* knockout embryos can develop to the late blastocyst stage when transcription defects start to manifest (Andreu et al, 2022). This discrepancy likely lies in the different methods of CTCF depletion. On the other hand, removal of cohesin derepresses minor ZGA genes in mESCs, and depletion of cohesin in donor cells facilitates minor ZGA and promotes SCNT development (Zhang et al, 2020a), indicating a loosened chromatin structure could promote ZGA. Moreover, overexpressing *Kdm4d*, an H3K9me3 demethylase, enhances the developmental potential of SCNT embryos. This enhancement is associated with the dissolution of H3K9me3-marked TADs, leading to the proposal that the weakened TADs may unleash a super-enhancer to activate *Zscan4* (Chen et al, 2020). It is currently unclear whether cohesin has a CTCF-independent role in gene regulation during ZGA.

Apart from loops, TADs, and compartments, other chromatin organizations also exhibit unique reprogramming in gametes and early embryos. Polycomb-associated domains (PADs) are H3K27me3-marked compartmental domains that appear in fully-grown mouse oocytes (FGOs) and on the maternal genome in mouse 1 to 8-cell embryos (Du et al, 2020). PADs may play a role in gene silencing in mouse FGOs (Du et al, 2020). Lamina-associated domains are undetected in oocytes, which correlates with activation of oocyte-specific enhancers (Liu et al, 2024), and are de novo established shortly after fertilization (Borsos et al, 2019). In 4-cell embryos and onwards, LADs colocalize with compartment B, which is preferentially associated with inactive genes and low gene densities, as in most other cell types. However, in late 2-cell embryos, LADs exist as a non-canonical state by showing weaker enrichment for compartment B. Unexpectedly, blocking ZGA using transcription inhibitors drives compartment A, which is usually associated with active genes and high gene densities, and regions marked by non-canonical H3K4me3 and H3K9ac, to LADs and the nuclear periphery (Borsos et al, 2019; Pal et al, 2023). Given that removing H3K4me3 in mouse zygotes eliminates LADs on the paternal allele (Borsos et al, 2019) and that non-canonical H3K4me3 has been proposed to be repressive (Dahl et al, 2016; Zhang et al, 2016), these data raise the possibility that non-canonical H3K4me3 could link active regions to the nuclear periphery, until ZGA corrects such pattern by establishing canonical LADs. How the reprogramming of these different chromatin organizations is intertwined with ZGA and their physiological relevance awaits further investigation.

## Licensors for ZGA

One fascinating question about ZGA is why major ZGA occurs precisely at a well-defined time point, such as 2-cell for mouse, 4-8-cell for human, and ~1000-cell for zebrafish (Schulz and Harrison, 2019). Whereas transcription factors can instruct activation of specific genes, it seems unlikely that they are primarily responsible for the timing of such synchronized expression for thousands of genes, often without apparent extrinsic timing signals. A clock mechanism driven by the accumulation/degradation of maternally-provided products may be at the center of control. Here, we discuss the emerging factors which do not have strong gene selectivity but are involved in the control of ZGA timing, and we call them "licensors" for ZGA. The concept of "licensing factors" was previously introduced for the pre-replication complex, which permits DNA replication by creating competent chromatin (Nishitani and Lygerou, 2002). Here, we refer to factors that enable the competency for transcription, which is the prerequisite for ZGA, as "licensors". Specifically, we define the "licensors" for ZGA as factors that meet the following criteria:

1. A licensor is essential to initiate ZGA; depleting this factor causes severe defects in ZGA.

2. A licensor exhibits a unidirectional change of abundance, maturity, or localization, etc. prior to ZGA and such changes generate a transcriptionally permissive state that underlies global transcription during ZGA. Preventing such changes impairs ZGA.

3. In principle, accelerating such changes should advance and promote ZGA. This effect however is also determined by the readiness of other licensing factors.

4. A licensor regulates global transcription without directly impacting on the selectivity of gene activation during ZGA.

It is worth noting that although epigenetic regulators often play essential roles in ZGA, they also come with certain gene specificities as individual epigenetic regulators often modulate a subset of ZGA genes with specific sequence features. For example, H3K4me3 is preferentially enriched at CpG-rich promoters (Deaton and Bird, 2011). Moreover, to be considered as "licensors", they would also need to meet the other criteria described above. For example, their nuclear or chromatin abundance is initially limited and is a bottleneck for ZGA, and changes in abundance later on permit ZGA to occur.

One general question before discussing licensors and specifiers is why major ZGA does not occur shortly after fertilization. In mouse early embryos, Pol II undergoes "loading", "pre-configuration" and "production" during the transition from minor to major ZGA (Abe et al, 2022; Liu et al, 2020). After dissociating from chromatin towards the end of oocyte growth, Pol II is re-loaded on chromatin ("loading") in 1-cell embryos, preferentially to high-CG promoters that include many developmental genes and housekeeping genes. Pol II then relocates to a group of low-CG promoters and dissociates from a subset of high-CG promoters, and is therefore ready for transcription elongation during ZGA ("pre-configuration") (Liu et al, 2020). Aberrant pre-configuration of Pol II, either by transient DRB treatment (Liu et al, 2020) or the knockout of the key ZGA factor *Obox* (Ji et al, 2023), leads to defective ZGA and ectopic activation of 1-cell Pol II targets. These data indicate that right after fertilization, a pre-ZGA period is required to correctly configure the transcription apparatus.

Two classic models have been proposed to explain the timing of ZGA: the nuclear to cytoplasmic ratio (N:C ratio) model and the maternal-clock model (Schulz and Harrison, 2019; Tadros and Lipshitz, 2009). The N:C ratio model proposes that the increasing N:C ratio titrates transcriptional repressors, leading to the initiation of ZGA. For example, decreasing the amount of DNA per cell by removing the maternal genome in embryos delays ZGA in frogs (Jukam et al, 2021). In both frogs and fish, the amount of core histones regulates ZGA (Amodeo et al, 2015; Joseph et al, 2017). In *Xenopus*, addition or depletion of H3/H4 decreased or increased transcription levels in the egg extract, respectively, and decreasing H3 levels induced premature transcriptional activation in vivo (Amodeo et al, 2015). In zebrafish, depleting histones advances ZGA from the middle 1k stage to the early 1k stage for a few examined genes, but whether this expression represents a genome-wide effect remains unknown (Joseph et al, 2017). It is also unclear whether such increased gene expression in the early 1k stage simply reflects a less transcriptionally-repressive chromatin structure. Increasing the N:C ratio in mouse embryos did not cause a significant change of the total transcription level but resulted in premature embryo compaction (Jukam et al, 2017). These differences may be related to much smaller changes in the N:C ratio from zygotes to 2-cells when mouse starts ZGA, unlike that in fast-developing animals (cell cycle 12-14 for *Xenopus*, zebrafish, and *Drosophila*).

The "maternal clock" model postulates a cell-cycle independent molecular timer. For example, the biochemical cascade initiated by fertilization or egg activation could gradually trigger the translation of maternally deposited mRNAs to ultimately initiate ZGA (Schulz and Harrison, 2019; Tadros and Lipshitz, 2009). The maternal clock model requires the translation of maternal mRNAs, including the transcription machinery, transcription activators, and maternal RNA clearance regulators (Fig. 3B). For example, transcripts of Zelda, a key ZGA regulator in *Drosophila*, were loaded in eggs and started to be translated upon egg activation (Larson et al, 2022). Smaug, a major regulator of maternal transcript destabilization in *Drosophila*, regulates ZGA through the destruction of mRNAs encoding the transcription repressor TTK (Benoit et al, 2009; Tadros et al, 2007). The translation of Zelda and Smaug is regulated by the Pan Gu (PNG) kinase, which is activated upon fertilization (Larson et al, 2022; Tadros et al, 2007).

Current evidence also points to a "nucleus gatekeeper" model, proposing that the nuclear-cytoplasmic localization of specific ZGA regulators is strictly controlled (Fig. 3C). General TF can recruit and assemble transcription machinery at promoters, whereas mediators or other coactivators can bridge the transcription machinery to sequence-specific TFs bound at enhancers (Wu and Vastenhouw, 2020). The cellular localization of the transcription machinery is highly regulated before and during ZGA. In nematodes, the transcription initiation factor TAF4 is released from the cytoplasm to the nucleus by degradation of its sequestering factor, the maternal proteins OMA1/2 (Vastenhouw et al, 2019). In zebrafish, nuclear pore complex (NPC) maturation may regulate the onset of ZGA (Shen et al, 2022). This process exerts control over the nuclear transport of maternal transcription factors, thereby influencing the precise timing of ZGA. Disruption of NPC maturation impairs the onset of ZGA. Overexpressing the NPC component *nup133* can promote ZGA (Shen et al, 2022). Collectively, these discoveries emphasize the importance of controlled nuclear protein import in orchestrating the timing of ZGA. The nuclear pore complex can be considered as a "licensor" for ZGA as its gradual maturation regulates the timing of ZGA and promotes ZGA without gene specificity.

In mouse zygotes, the Pol II C-terminal domain gradually acquires S2 phosphorylation, which is required for elongation by Pol II (Oqani et al, 2011). Components of the transcription elongation factor-b (p-TEFb) complex, which phosphorylates the C-terminal domain of Pol II, are initially cytoplasmic but translocate into the nucleus at the late 1-cell or 2-cell stage in mouse (Liu et al, 2020; Oqani et al, 2011). Treatment with the CDK inhibitor flavopiridol results in mislocalization of p-TEFb components CDK9, Cyclin T1 and phosphorylated Pol II, as well as developmental arrest at the 2-cell stage (Oqani et al, 2011). Furthermore, Cyclin T2, another component of p-TEFb, is present in the nucleus already at the 1-cell stage. Knocking down cyclin T2 in GV oocytes resulted in a significant number of embryos arrested at the 2- or 4-cell stage and downregulation of 12.8% of mouse ZGA genes (Zhang et al, 2022). TDP-43, an RNA-binding protein, also regulates the p-TEFb complex and ZGA. Depletion of maternal TDP-43 in mouse oocytes downregulated ~23.7% of major ZGA genes and led to embryo arrest at the 2-cell stage (Nie et al, 2023). During mouse ZGA,

A. Titration/removal of repressors

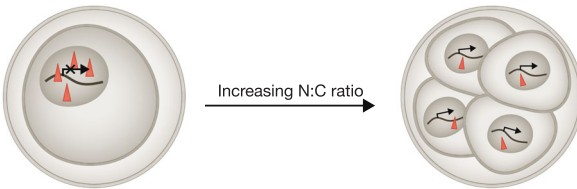

Increasing N:C ratio

B. Accumulation of activators

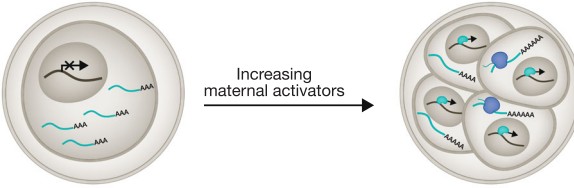

Increasing
maternal activators

C. "Nucleus gating" model

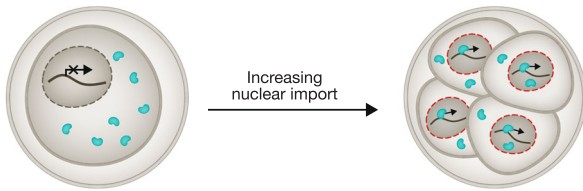

Increasing
nuclear import

**Figure 3.   Models of ZGA timing control.**

**(A)** "Titration/removal of repressors" model. The dilution of transcriptional repressors (red triangles) by DNA replication and cell division initiates the activation of ZGA genes. **(B)** "Accumulation of activators" model. Active factors (in green) are gradually translated (by ribosomes in blue) after fertilization and act as a molecular timer to trigger ZGA. **(C)** "Nucleus gating" model. The maturation of nuclear import pathways facilitates the entry of key regulators (in green) into the nucleus, which activates ZGA.

TDP-43 translocates from the cytoplasm to the nucleus. TDP-43 interacts with Pol II and Cyclin T1 within the p-TEFb complex. Intriguingly, TDP-43 deficiency results in the loss of Pol II S2 phosphorylation condensates (Nie et al, 2023). These data suggest that TDP-43 plays a regulatory role in Pol II configuration and ZGA gene expression. Factors like Cyclin T1 that exhibit increasing nuclear levels prior to ZGA can also be regarded as "licensors" for ZGA.

## Specifiers (transcription factors) of ZGA

TFs are key drivers of embryo-specific programs. At least three criteria were proposed to identify potential key transcription factors for ZGA (Wu and Vastenhouw, 2020). First, they should be highly expressed and accumulated prior to the major wave of ZGA. Some key maternal TFs may undergo translational upregulation during oocyte-to-embryo transition (or maternal-to-zygote transition, MZT). Second, their DNA-binding motifs should be highly enriched in the promoter or enhancer of ZGA genes to enable expression of hundreds or thousands of zygotic genes at a time. Third, depletion of these TFs should lead to developmental defects and mis-regulation of ZGA genes in early embryos. The identity of these TFs has been a long-standing mystery in most species with a few exceptions (discussed below). Recently, identification of

candidate key TFs has been greatly accelerated by low-input transcriptome and translatome analyses, which uncovered highly expressed TFs prior to and during ZGA, as well as TF footprint analyses and low-input chromatin analysis technologies, including low-input ATAC-seq (Jung et al, 2019; Jung et al, 2017; Wu et al, 2016; Wu et al, 2018), DNase-seq (Gao et al, 2018; Lu et al, 2016), MNase-seq (Wang et al, 2022a), scCOOL-seq (Guo et al, 2017; Li et al, 2018a). These low-input chromatin analyses enable researchers to identify active promoters and enhancers near ZGA genes and the DNA-binding motifs implicated in TF binding. Here we review the current knowledge related to key TFs for ZGA in zebrafish, fly, mouse, and human, and the molecular mechanisms underlying their actions.

### Drosophila

In *Drosophila*, minor ZGA takes place at the mitotic cleavage cycles (nuclear cycle, or NC) 8-13, followed by the major wave of ZGA at cleavage cycle 14 (NC14) (Liang et al, 2008)) (see also the review in this issue of EMBO reports by Ciabrelli et al) (Ciabrelli et al, 2024). The heptamer "CAGG<u>TAG</u>" together with its two 1 bp-degenerate sequences, collectively named as TAGteam, is over-represented in the upstream regulatory regions of early transcribed zygotic genes (ten Bosch et al, 2006). Zinc-finger protein Zelda (Zld) was identified in a screen as the potential binding factor for the TAGteam motif and as the first key zygotic genome activator in *Drosophila* (Liang et al, 2008; Zhu et al, 2011). Although high levels of maternal *Zld* transcripts were loaded in ovaries and eggs, these transcripts were not translated until 1 h after fertilization, due to the relief of translational repression by the RNA-binding protein Brain tumor (BRAT) (Larson et al, 2022). Prior to major ZGA, Zld already binds to thousands of cis-regulatory regions to establish or maintain open chromatin as a pioneer factor (Harrison and Eisen, 2015). Depletion of *Zld* led to developmental lethality and downregulation of hundreds of early zygotic genes. The majority of these downregulated zygotic genes have TAGteam sites nearby (Liang et al, 2008; McDaniel et al, 2019).

Apart from TAGteam, GA-dinucleotides, which are also found enriched in accessible regions during ZGA in the absence of Zld, are bound by two other maternal TFs, GAGA Factor (GAF) and Chromatin-linked adaptor for MSL proteins (CLAMP), especially at promoters (Duan et al, 2021; Gaskill et al, 2021; Soruco et al, 2013). GAF and CLAMP were also required for chromatin accessibility and gene activation during ZGA (Duan et al, 2021; Gaskill et al, 2021). GAF and CLAMP have overlapping but not identical roles, and they compete for a subset of binding sites (Kaye et al, 2018). They and Zld largely function independently but bind cooperatively in some regions. As a result, zygotic genes can be classified as ZLD-dependent, GAF-dependent, and CLAMP-dependent based on whether their expression and promoter accessibility are affected by the depletion of each TF (Duan et al, 2021; Gaskill et al, 2021).

### *Zebrafish*

In zebrafish, ZGA occurs during the midblastula transition (MBT) around 3 hpf (3 h post-fertilization) (Lee et al, 2013). The three maternal transcripts of Nanog, Pou5f3 (also named Pou5f1), and Sox19b (or SoxB1 family) (NPS) encode the most highly translated TFs prior to ZGA, which play important roles in ZGA (Belting et al,

2001; Lee et al, 2013; Leichsenring et al, 2013; Okuda et al, 2010; Okuda et al, 2006; Onichtchouk et al, 2010; Xu et al, 2012) at the pre-MBT stage. Pou5f3 binds to SOX-POU binding sites, which preferentially reside near ZGA genes (Leichsenring et al, 2013). Depleting Pou5f3, Nanog, and Sox19b in zebrafish strongly inhibited ZGA and embryonic development in an additive, combinatorial, and cooperative manner for different sets of genes (Lee et al, 2013; Miao et al, 2022). Without NPS, ~37% (822 out of 2240) of zygotic genes were downregulated, with a strong reduction of H3K4me3 and H3K27ac at promoters, and H3K4me1 and H3K27ac at enhancers of these downregulated genes. NPS preferentially controls the establishment of enhancers, as the accessibility of over 50% of active enhancers and only 5.1% of active promoters is NPS-dependent. NPS is required to recruit histone acetyltransferase p300 and a histone acetylation reader BRD4 to enhancers and some promoters, and artificial recruitment of the core domain of p300 can bypass the requirement for TFs to activate genes (Miao et al, 2022). These data demonstrate that master TFs function through histone acetyltransferases to activate specific genes during ZGA.

### Mouse

In mouse, minor ZGA occurs around the late 1-cell and early 2-cell stage (Jukam et al, 2017; Lee et al, 2014), giving rise to dozens of genes showing high expression as well as promiscuous, low levels of transcription genome-wide (Abe et al, 2015). Major ZGA occurs during the late 2-cell stage (Jukam et al, 2017; Lee et al, 2014). Dux in mouse (DUX4 in human) is a double homeodomain TF widely present in placental mammals. DUX was identified as a candidate TF for ZGA as it is expressed during minor ZGA and its binding motif is enriched in promoters of genes that are expressed during ZGA in cleavage-stage embryos (De Iaco et al, 2017; Hendrickson et al, 2017; Whiddon et al, 2017). Ectopic expression of mouse DUX or human DUX4 in ESCs can activate a subset of ZGA genes, with many being minor ZGA genes (for example, *Zscan4*) (De Iaco et al, 2017; Hendrickson et al, 2017; Whiddon et al, 2017). DUX also binds and activates transposable elements including MERVL/ HERVL and ERVL–MaLR retrotransposons (Alabert et al, 2014; De Iaco et al, 2017; Geng et al, 2012; Hendrickson et al, 2017). However, *Dux* homozygous knockout mice only exhibit relatively minor defects in ZGA and their developmental potential, and can survive to adulthood although with reduced litter sizes, indicating that DUX is not strictly required for ZGA and embryonic development (Bosnakovski et al, 2021; Chen and Zhang, 2019; De Iaco et al, 2020; Guo et al, 2019).

Two maternal proteins DPPA2 and DPPA4 directly bind to the *Dux* promoter and gene body to drive mouse *Dux* expression and the 2C-like program in mESCs (De Iaco et al, 2019; Eckersley-Maslin et al, 2019; Yan et al, 2019). However, both *Dppa2/4* maternal knockout or maternal-zygotic knockout mice are viable with little impact on ZGA (Chen et al, 2021b; Kubinyecz et al, 2021). At the 2-cell stage, the motif of NFYA is enriched in the promoter DNase I-hypersensitive sites (DHSs) (which indicate accessible regions). Knockdown of maternal *Nfya* led to embryo arrest at the morula stage, accompanied by the downregulation of 15.1% (297 out of 1961) of ZGA genes and reduction of accessibility for 28.4% of 2-cell DHSs especially at promoter regions around downregulated genes (Lu et al, 2016). The transcription of NR5A2, a nuclear receptor TF suggested to have lost ligand binding ability

in mouse (Giguère, 1999; Krylova et al, 2005), is strongly induced during ZGA, noting a low abundance of the maternal transcript (Gassler et al, 2022). NR5A2 was proposed to be required for ZGA and development beyond the 2-cell stage, because embryos treated with the nuclear receptor inhibitor SR1848 exhibited downregulation of 72% ZGA genes and 2-cell arrest (Gassler et al, 2022). However, two other research groups found NR5A2 primarily regulates the 4-cell and 8-cell, but not 2-cell transcription programs using *Nr5a2* KD or KO, with a moderate effect on ZGA (1.6-3% ZGA genes downregulated) (Festuccia et al, 2023; Lai et al, 2023; Wu et al, 2016). *Nr5a2* maternal-zygotic KO (mzKO) or *Nr5a2* KD led to developmental abnormalities around the morula stage (Festuccia et al, 2023; Lai et al, 2023). Consistently, depletion of *Nr5a2* decreased chromatin accessibility in 8-cell embryos but not in 2-cell embryos (Lai et al, 2023), indicating NR5A2 is likely not a major regulator for mouse ZGA but functions primarily at the 4–8 cell stages.

More recently, the OBOX proteins, a group of rodent-specific TFs exclusively expressed in oocytes and early embryos (Royall et al, 2018), have been found to be essential for mouse ZGA (Ji et al, 2023; Sakamoto et al, 2024); OBOX binding motifs are enriched at promoters and enhancers near 2-cell specific genes in mouse (Ge, 2017; Ji et al, 2023). The OBOX family includes maternally expressed *Obox1/2/5/7*, minor ZGA-expressed *Obox4* and its pseudogenes, and major ZGA-expressed *Obox3/6/8*. Obox1/3/6 could activate ZGA genes when ectopically expressed in embryonic fibroblasts (Royall et al, 2018). Maternally expressed *Obox1/2/5/7* contain proximal cytoplasmic polyadenylation elements (papCPEs) in their 3' UTRs (3' untranslated regions) (Ji et al, 2023), a signature of "dormant RNAs" that are transcribed during oocyte growth but remain translationally inactive until meiotic resumption, when they are recruited to polysomes for translation (Dai et al, 2019; Richter, 2007; Xiong et al, 2022). A recent study showed that the knockout of maternally expressed *Obox1/2/5/7* and embryonically-expressed *Obox3/4* in mice (by deletion of a 1.2 Mb region containing this gene cluster) resulted in 2-4 cell arrest and severe ZGA defects for 32% of minor ZGA genes (e.g., *Zscan4d*) and 48% of major ZGA genes (e.g., *Dppa2* and *Nr5a2*) (Ji et al, 2023). The developmental and ZGA defects could be rescued by overexpression of either maternal *Obox1/5/7* or zygotic *Obox* (*Obox3*), indicating they function redundantly in mouse ZGA and early development. Ectopic expression of OBOX in mESCs activated a subset of ZGA genes and *MERVL* (Ji et al, 2023; Sakamoto et al, 2024). In mouse 2-cell embryos, OBOX binds and regulates chromatin opening at promoters and enhancers of ZGA genes. Importantly, OBOX also guides the recruitment of Pol II to ZGA genes. OBOX is required for the timely pre-configuration of Pol II during ZGA because it binds prior to Pol II and helps recruit Pol II to CG-poor 2-cell-specific binding sites. *Obox* mzKO led to aberrant Pol II retention on 1-cell-specific sites in WT embryos and failed recruitment to ZGA genes. As a result, both aberrant activation of ZGA genes and ectopic activation of Pol II 1-cell targets were found in these mutant embryos (Ji et al, 2023). OBOX and DUX appear to independently control different sets of ZGA genes, with OBOX regulating both minor and major ZGA genes whereas DUX regulates primarily minor ZGA genes (Ji et al, 2023) (Fig. 4). Of note, a recent study reported that whereas individual knockdown of *Obox4* and *Dux* caused minor defects in embryogenesis, *Obox4/Dux* double knockdown impaired

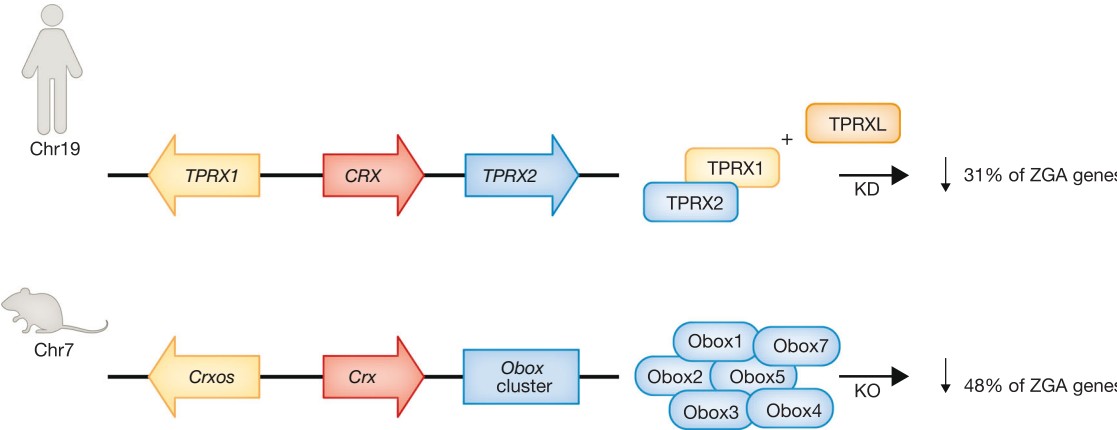

**Figure 4. TPRXs and OBOXs function in human and mouse ZGA, respectively.**

Human *TPRX1/2* genes and mouse *Croxs/Obox* genes originated from the duplication of the Otx family gene *Crx* and both belong to the Eutherian Totipotent Cell Homeobox (ETCHbox) genes (Royall et al, 2018). Mouse *Croxs/Obox* are highly divergent orthologues of *TPRX1* and *TPRX2*, respectively. *TPRXL* was likely generated by reverse transcription of TPRX1 mRNA followed by genome integration (Booth and Holland, 2007). *TPRX1/2/L* triple knockdown in human 3 pronuclei (3PN) embryos and *Obox1/2/3/4/5/7* maternal-zygotic KO in mouse downregulated about 31% and 48% of human and mouse ZGA genes (Ji et al, 2023; Zou et al, 2022), respectively, indicating their conserved roles in ZGA.

development to the blastocyst stage (Guo et al, 2024), suggesting that they may have compensatory functions in ZGA. Interestingly, the expression of *Obox* and *Dux* is both impaired in SCNT embryos (Sakamoto et al, 2024). Consistently, restoration of *Obox3* or *Dux* can promote the efficiency of SCNT (Sakamoto et al, 2024), supporting roles for OBOX and DUX in totipotency acquisition.

*Human*

In human, a major wave of ZGA occurs around the 4-8-cell stage (Jukam et al, 2017; Lee et al, 2014). Searching for key TFs for human ZGA has been a long-standing challenge due to the limited research materials from human embryos. Expressing *DUX4* in human embryonic stem cells (hESCs) activated signature genes of 8-cell-like cells (8CLCs) and increased the 8CLC population among naïve hESCs (Taubenschmid-Stowers et al, 2022; Yu et al, 2022b). The expression of DUX4 also led to an increase of ACRO1 and HSATII satellite repeat transcripts that are enriched in cleavage embryos (Hendrickson et al, 2017). DUX4 can be activated by p53 in ESCs and FSHD (Facioscapulohumeral muscular dystrophy) cell models (Grow et al, 2021). P300/CBP and MED15 were identified as DUX4-interacting partners to help activate transcription and remodel chromatin, respectively (Choi et al, 2016; Vuoristo et al, 2022). However, *DUX4* knockdown in human early embryos only led to minor changes in the activation of ZGA genes (Liu et al, 2022b; Vuoristo et al, 2022). In addition, the POU5F1/OCT4 DNA-binding motif was found to be moderately enriched in DHS (DNase I hypersensitive sites) of human 8-cell embryos (Gao et al, 2018). Knockdown of OCT4 in human zygotes resulted in the downregulation of 25% of ZGA genes (Gao et al, 2018). On the other hand, OCT4 proteins are not detected until the 8-cell stage (Niakan and Eggan, 2013) and knockout of OCT4 by CRISPR-Cas9 appears to have a late-stage impact including failure of blastocyst development and lack of expression of genes associated with lineage specification (Fogarty et al, 2017). Therefore, the roles of DUX and OCT4 in human ZGA remain to be further elucidated in the future.

An Alu element, also named "EGA-enriched Alu-motif, EEA motif", harbors a core "TAATCC" motif that is over-represented in the promoters of human ZGA genes (Ge, 2017; Katayama et al, 2018). This motif also stands out in ATAC-seq peaks in human 8-cell embryos (Wu et al, 2018; Zou et al, 2022). A group of PRD-like homeobox TFs bind this motif and activate reporters, including OTX2, LEUTX, TPRX1, TPRX2, and DPRX, which share a similar K50-type homeobox binding domain (the PRD-like family is characterized by distinct residues at position 50 of the homeodomain) (Katayama et al, 2018). Overexpression of these TFs in hESCs activates a subset of ZGA genes (Madissoon et al, 2016). TPRX1 was further identified as a marker of 8CLCs (Mazid et al, 2022; Taubenschmid-Stowers et al, 2022; Yu et al, 2022b). Moreover, OTX2, LEUTX, TPRX1, TPRX2, and a TPRX1-related TPRXL protein are among the most highly translated TFs prior to major ZGA (Zou et al, 2022). OTX2 and TPRXL both contain papCPEs, suggesting that they are dormant RNAs (Zou et al, 2022). TPRX1/2 seem to be redundant in human embryos, as individual knockdown yielded much smaller defects in transcription than combined TPRX1/2 knockdown (Zou et al, 2022). Triple knockdown of *TPRX1/2/L* genes led to defective activation of 31% of ZGA genes, with most embryos arrested at the 4-9 cell stage (Fig. 4), and the downregulated ZGA genes were enriched for TPRX binding sites. Affected ZGA genes include key genes such as *ZSCAN4*, other PRD-like homeobox genes *DUXA*, *DUXB*, *DPRX*, *ARGFX*, and TFs potentially related to lineage commitment, *DPPA4*, *GATA6*, and *KLF5*. Some of these genes are also bound and activated by ectopically expressed TPRX1 or TPRX2 in hESCs. How these TFs cooperatively regulate ZGA and whether they work together with other TFs remain to be further explored. It should be noted that TPRX1/2 in human and the OBOX family in mouse both arose by gene duplication from the retinal-expressed Crx gene (Maeso et al, 2016), indicating their conserved functions in human and mouse ZGA (Fig. 4).

## The interplay among transposable elements, key TFs, and ZGA

Transposable elements are active in early development (Garcia-Perez et al, 2016; Gifford et al, 2013). Accumulating evidence suggests that their activation is not simply a consequence of epigenetic remodeling during embryonic development. Instead, these activated elements re-wire the transcription regulatory circuitry through various ways including by serving as enhancers and promoters for endogenous genes and by modulating global chromatin states to regulate embryonic transcription programs. For example, the transcription of MERVL and HERVL, members of endogenous retroviruses (ERVs), can be activated by DUX in mouse and DUX4 in human, respectively (Hendrickson et al, 2017). MERVL and HERVL, as LTR retrotransposon, have LTRs on both ends, which serve as the promoters and terminators of their expression. The transcription activation of MERVL in turn activates hundreds of ZGA genes in 2-cell-like cells (2CLCs), in part by functioning as alternative promoters to generate chimeric transcripts (Genet and Torres-Padilla, 2020; Macfarlan et al, 2012; Peaston et al, 2004). Knockdown and CRISPRi-based repression of MERVL transcription results in embryonic lethality and aberrant expression of 2-cell-specific genes, suggesting MERVL plays an important role in pre-implantation embryo development (Kigami et al, 2003; Sakashita et al, 2023). The expression of MT2_Mm, the LTR of MERVL, was found to be activated by OBOX and DUX, and MT2_Mm has been proposed to function as promoters/enhancers to regulate ZGA (Ji et al, 2023; Yang et al, 2024). Perturbation of MT2_Mm using CRISPRi leads to downregulation of hundreds of ZGA genes and embryonic arrest at the morula stage (Yang et al, 2024). LINE-1 is one of the most abundant retrotransposon families in mammals and is transcribed soon after fertilization (Fadloun et al, 2013; Jachowicz et al, 2017). Premature silencing of LINE-1 decreases chromatin accessibility at the 2-cell stage, whereas prolonged activation prevents the gradual chromatin compaction at the 8-cell stage (Jachowicz et al, 2017). Interestingly, knocking down LINE-1 also causes a prolonged minor ZGA program and 2-cell arrest (Percharde et al, 2018). A similar event was observed in ESCs where LINE1 knockdown activates 2-cell state genes or minor ZGA genes. This activation is in part achieved through the partnership between the LINE1 transcript and nucleolin/KAP1 to repress Dux through peri-nucleolar heterochromatin (Percharde et al, 2018; Xie et al, 2022; Yu et al, 2021). Overall, these data point to a dual role of LINE1 in both promoting chromatin accessibility and silencing the minor ZGA program. Such seemingly contrasting actions may be due to different stages and contexts.

B1 elements in mouse and Alu elements in human are the most prevalent retrotransposons of Short Interspersed Nuclear Elements (SINEs), covering 3-11% of mouse and human genomes (Lu et al, 2021a). B1 and Alu elements are transiently transcribed and become accessible (assayed by ATAC-seq) in mouse 2-cell embryos and human 8-cell embryos, and harbor binding motifs for key ZGA factors (Lu et al, 2021a; Wu et al, 2016; Wu et al, 2018). A "super motif" consisting of six binding motifs, including NR5A2 and OBOX, was found upstream of 70–77% of ZGA genes and embedded in SINE B1 retrotransposons (Gassler et al, 2022; Ji et al, 2023; Lai et al, 2023). OBOX preferentially binds B1/B2/B4 elements and chromatin becomes less accessible in OBOX KO 2-cell embryos (Ji et al, 2023). NR5A2, by contrast, has a minimal effect on chromatin accessibility at the 2-cell stage but has a greater

impact at the 8-cell stage (Lai et al, 2023). One interesting possibility is that OBOX may bind and open B1 elements, which then allow the binding of other TFs such as NR5A2. These data support the idea that transcription of transposable elements may help shape and maintain chromatin accessibility to regulate gene activation in their neighborhood. Currently, it remains unclear why SINE is abundant near ZGA genes, but it is tempting to postulate that mammals evolved such a way to efficiently coordinate gene expression using a minimum number of transcription factors by utilizing the prevalent repeat elements in the genome to trigger widespread ZGA in early embryos. Alternatively, these repeats may evolve and converge to sequences that contain motifs of the master TFs.

Box 1 In need of answers

Going forward, many burning questions need to be answered to fully understand the mechanisms underlying ZGA. We listed a few examples below.

1. What are other key ZGA TFs in mammals? How do different ZGA TFs cooperatively activate ZGA? Thus far, knockout of a master ZGA TF usually affects fewer than half of ZGA genes, suggesting the presence of additional key ZGA TFs. How do they cooperate to initiate the complete ZGA program?

2. What are the key ZGA TFs in other species? While OBOX and TPRX belong to the same TF family (PRDL homeobox TFs), why do ZGA TFs in non-mammalian species seem to diverge substantially from those in mammals? For example, distinct ZGA TFs were identified in flies (Zelda, GAF, and CLAMP), zebrafish (Nanog, Pou5f1, and SoxB1), and mammals (DUX, OBOX, and TPRX). The principle and logic underlying such divergence are still elusive.

3. How do TFs and chromatin regulators cooperatively specify ZGA genes? Early embryos have highly dynamic, non-canonical chromatin environments. How to initiate ZGA by overcoming epigenetic barriers and epigenetic instability caused by genome-wide epigenetic resetting remains an intriguing question.

4. What controls the timing of ZGA? ZGA timing is likely regulated by multiple pathways and "licensors". Identification of these factors ideally requires evidence from both loss-of-function and gain-of-function experiments. Nevertheless, evidence showing that ZGA timing could be advanced by overexpression of a ZGA regulator is still rare, possibly due to the presence of redundant licensors.

5. Although various cell models have been developed (such as 2C-like cells) to model ZGA, many "2C genes" activated in these cells are expressed during minor ZGA (e.g., Dux, Zscan4, MERVL, etc.). Thus far, major ZGA still lacks an effective in vitro cell model. Establishing these long-needed models for biochemical analyses and genetic screening will be highly desirable to unveil the fundamental nature of ZGA.

## Conclusion

The last decade has witnessed rapid progress toward understanding the molecular nature of ZGA. In this review, we discussed how ZGA "licensors", including regulators of the transcription apparatus and nuclear gatekeepers, create a permissive state for ZGA, whereas key TFs then serve as "specifiers" to instruct subsets of genes to be activated. The latest findings have raised more questions that need to be answered (Box 1), including those that involve the identities of key regulators of ZGA, their action mechanisms, and their conservation and species-specific

innovation during evolution. Leveraging low-input and single-cell technology will be highly desirable to address these fascinating questions of ZGA. On the other hand, lack of accessible tools of genetic screening and biochemical analysis in early embryos however is still a major bottleneck for identifying new factors controlling ZGA. Novel in vitro models that closely mimic early-stage mammalian embryos and low-input proteome technology are expected to greatly accelerate the pace of discovery. Finally, application of this knowledge to improve IVFs and treat infertility will be a crucial step to bring these findings to clinics.

## Peer review information

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

## Acknowledgements

We thank the laboratory members for the helpful discussion and feedback, particularly Yao Wang, and Bofeng Liu during the preparation of the manuscript. The review included selected studies due to the space limitation, and we sincerely apologize to all the authors whose work was not cited. This work was funded by the National Natural Science Foundation of China (31830047 and 31988101 to WX), the National Key R&D Program of China (2021YFA1100102 and 2019YFA0508900 to WX), and the Tsinghua-Peking Center for Life Sciences (WX). WX is a recipient of an HHMI International Research Scholar award and is a New Cornerstone Investigator.

## Author contributions

**Zhuoning Zou**: Conceptualization; Visualization; Writing—original draft; Writing—review and editing. **Qiuyan Wang**: Conceptualization; Visualization; Writing—original draft; Writing—review and editing. **Xi Wu**: Writing—original draft. **Richard M Schultz**: Writing—review and editing. **Wei Xie**:

Conceptualization; Supervision; Funding acquisition; Writing—original draft; Project administration; Writing—review and editing.

## Disclosure and competing interests statement

The authors declare no competing interests.

