## [Peer Review File · EMBO Reports]

Kick-starting the zygotic genome: licensors, specifiers, and beyond

Zhuoning Zou, Qiuyan Wang, Xi Wu, Richard M. Schultz, and Wei Xie

Corresponding author(s): Wei Xie (xiewei121@tsinghua.edu.cn)

Review Timeline:

Submission Date:	10th Feb 24
Editorial Decision:	22nd Apr 24
Revision Received:	14th Jun 24
Accepted:	24th Jul 24

Editor: *Esther Schnapp*

Transaction Report:

Dear Wei,

Thank you for your patience while your review was assessed by referees at EMBO reports. Below you will find the set of comments we have received on your manuscript. Unfortunately, referee 2 has still not sent his/her report and is unresponsive to our chasers. I have therefore contacted another advisor for some more comments on your review, which are also included below.

As you will see, all referees acknowledge that the review covers an interesting topic. While referee 1 feels that calling the ZGA regulators licensors and specifiers is not helpful, referee 3 and the advisor find this an interesting concept and suggest how it could be expanded. I am fine with you to decide whose suggestion to follow. The referees also point out missing and incorrect references, and referee 1 further raises more points that I think should all be addressed, if you agree, of course.

I would thus like to invite you to revise your review along the lines suggested by the referees. Please co-submit with your revised review a detailed point-by-point response to facilitate its evaluation.

Please also add a Box called "In need of answers" to the review that lists open questions in the field, which can be accompanied by suggested experiments for how to address these.

I think the review figures are good as they are and do not need to be re-drawn by a graphics designer, which would also delay the publication process by at least 10 working days. Please let me know in case you disagree. If BioRender was used to generate the figures, this must be cited either in the figure legends or mentioned in the acknowledgement section.

As for timing, would it be possible for you to submit the revised review by May 10th? If you anticipate a problem meeting this date, then please just let me know.

Thank you again for contributing this nice piece to EMBO reports.

With best wishes,

Esther

Referee #1:

This is a largely comprehensive, well-written review on the interesting topic of zygotic genome activation in embryos. The authors provide the reader with a broad view of the field and information on the epigenetic changes, ZGA models and transcription factors. The greatest strength of the review is its breadth. The main weakness is that the review includes inaccuracies of the literature and omissions of relevant work. These need to be corrected before the review can be considered appropriate for publication.

In addition, the authors classify ZGA regulators into the categories of licensors and specifiers. While conceptual groupings can be helpful to simplify complex processes, the proposed use of "licensors" in this context falls short of improving the understanding of ZGA for several reasons. First, licensors apparently can be anything - components of transcriptional machinery, epigenetic regulators, metabolic regulators, nuclear pore complex - except transcription factors, which are classified as "specifiers". This means that molecules involved in any molecular events that precede transcription factor binding and RNA pol II activity are licensors - this must necessarily include nuclear assembly, chromatin assembly and even the ribosomes to facilitate translation. Therefore, the term "licensor" has no molecular meaning. Secondly, licensing as a concept has been used previously to understand the once-per-cell-cycle regulation of DNA replication. DNA replication licensing can occur only under conditions of low cyclin-dependent kinase activity and the licensing pathway was later identified as the assembly of the pre-replication complex. I would therefore argue that once the identities of proteins involved are known, the term of "licensing" becomes obsolete and one should refer to the actual protein complexes and mechanisms. Therefore, I do not think that introducing "specifiers" and "licensors" at the current time are helpful for a better understanding of ZGA.

This review overlaps to some extent with Zhang & Wei, *Curr Opin Genet Dev*, "Building the genome architecture during the maternal to zygotic transition" and Du et al. Wei, *Cold Spring Harb Perspect Biol* 2022, "Epigenetic Reprogramming in Early Animal Development".

Main points:

1. The schematic of 3D chromatin (Fig 2) is inaccurate for both maternal and paternal genomes and requires extensive revision. 3D genome organization comprises loops, TADs and compartments. Given that loops/TADs and compartments are antagonistic in many cell types, it is unclear how these different aspects of 3D chromatin can be summarized as one category. Regarding maternal chromatin: 1) MII oocytes lack loops and TADs since these are condensed chromosomes, as had been shown previously for mitotic cells (Naumova et al., Science 2013). 2) Loops and TADs are present in maternal and paternal chromatin of one-cell embryos (Flyamer et al., Nature 2017) and both are dependent on cohesin in one-cell embryos (Gassler et al., EMBO J 2017). Regarding paternal chromatin: Although several publications including by the authors of this review have reported the presence of TADs and compartments in mouse sperm, the work by Yin et al., Genome Research 2023 has convincingly demonstrated that the source of these 3D genome features is cell-free DNA in sperm preparations. The authors point out that TADs and compartments in sperm are under debate in the figure legend, but actually the issue has been settled and this should be accurately reflected in the schematic and also openly discussed in the main text.
2. An interesting aspect of early genome architecture is that it appears to be more relaxed and the authors discuss whether this could be related to expression of CTCF (lines 305-308). It is stated that the expression of CTCF during development cannot explain the genome architecture change during ZGA of the fly. However, this does not make sense anyhow as CTCF is known not to be present at fly domain boundaries. Therefore, fly should not be discussed in this context. Instead, it should be pointed out that CTCF is insulating TAD boundaries in 1-cell embryos as has been demonstrated by Hi-C of oocyte-specific genetic knockdown of CTCF (Dequeker et al., Nature 2022). Therefore, CTCF is constraining loop extrusion in early embryos.
3. The authors describe a recent publication (Andreu et al., 2022) that uses a conditional genetic knockout approach to delete CTCF in oocytes and claim that CTCF-null embryos have largely unaffected transcription at the morula stage and develop to the blastocyst stage. CTCF, and especially the chromatin-bound fraction, is notoriously difficult to deplete. The paper does not provide any evidence that the conditional genetic knockout approach causes depletion of CTCF protein in oocytes or early embryos prior to the morula stage. It is therefore unclear whether the lack of phenotype could be due to the lack of efficient protein depletion. Indeed, the results of this paper contradict earlier work by Marisa Bartolomei. Using a genetically encoded oocyte-specific dsRNAi targeting CTCF, it was shown that CTCF loss affects gene expression in oocytes, causes defects in zygotic gene expression and an embryonic arrest around the 2- to 4-cell stage (Wan et al., Development 2008). It would be more appropriate to discuss this work in the paragraph dedicated to how genome architecture and ZGA might be related.
4. It is surprising that the authors repeatedly fail to reference the first published mouse embryo Hi-C paper (Flyamer et al., Nature 2017), when describing the 3D genome in embryos (lines 300, 302). The review should be scholarly and reference this work in both instances.
5. The authors describe the data linking DNA methylation to ZGA. They state that "gametic methylation... is not expected to have a major impact on ZGA, as Dnmt3a or Dnmt3l maternal knockout (mKO) embryos, which lose all DNA methylation in oocytes, can develop beyond the pre-implantation period" (lines 96-99), yet the paragraph concludes that "proper DNA methylation reprogramming is critical for the fidelity of ZGA" (lines 113-114). These statements contradict each other. The paragraph ending needs to be toned down to reflect that DNA methylation reprogramming can promote the fidelity of ZGA.
6. The claim is made that "histone acetylation and deacetylation are both essential for mouse ZGA and preimplantation development" (lines 142-143). The perturbations in embryos relied on expression of dominant negatives or inhibitors and therefore indirect effects cannot be excluded. In cultured cells, the genetic engineering of 28 alleles of H3 to replace lysine 27 with arginine showed that H3K27 acetylation is not essential for gene activation, at least in the transition to epiblast-like cells (Sankar et al., Nat Genet 2022). The embryo data should be discussed in light of these findings and the conclusion quoted above needs to be toned down.
7. The authors should point out in the main text that the "knockout of maternally-expressed Obox1/2/5/7 and embryonically-expressed Obox3/4" (lines 520-521) was achieved by deletion of a 1.2 Mb region containing these and other pseudogenes.
8. Towards the middle of the review, the authors start discussing work in first person, e.g. "we showed that..." (line 520). This is principally of course fine and the authors are at liberty to do so. However, it seems odd that at this point a switch to first person occurs whereas many of the other referenced work earlier was also from this group. It would be good to be consistent throughout the review.

Referee #3:

The authors summarized the latest findings about zygotic genome activation (ZGA), which is of broad interest. Although the ZGA has long been known, the mechanisms controlling ZGA have remained unclear. In recent years, new factors involved in the ZGA such as Dux and Obox have been found, and therefore this review about ZGA is timely and important. In addition, the authors categorize the ZGA-regulating factors into Licensors and Specifiers, enhancing the understandability of the text. The various

aspects discussed in the review are relevant to a broad audience. I have a few suggestions to further improve the manuscript.

Comments

1.107-109

The sentence flow around "However" is confusing and should be revised to increase clarity.

2.134

Fig. 1b is not present in the text. Please correct it.

3.150

The cited reference (Chen et al. 2014 "Single-molecule dynamics of enhanceosome assembly in embryonic stem cells.") did not match the text.

4.166-169

It has been reported that the switch from non-canonical to canonical H3K4me3 via KDM5 is important for the early development of bovine and porcine embryos (Dang et al. 2022, Bu et al. 2022), which should be included.

5.199-200

Citations are needed for these reports.

6. 267-268

There are no references about Gap-filling of H3 variants.

7. 274-276

As far as I know, H3.1/3.2, which are canonical in mammals, are DNA replication dependent. It is unclear why fly oocytes can live with canonical H3 without DNA replication. If possible, please add an explanation regarding this.

8.757-760

There are duplicate citations.

9.787-788

This paper has already been published in Elife. The information of journal should be corrected.

10.833-836

There are duplicate citations.

11.857-862

There are duplicate citations.

12.1093-1094

This paper has already been published in Genome Research. The information of journal should be corrected.

13.1103-1108

There are duplicate citations.

Advisor's comments:

This review by Zou and colleagues, provides an excellent summary of the recent progress concerning the regulation of zygotic genome activation across species, from a transcriptional and epigenetic perspective. It provides a timely update of current knowledge in a rapidly advancing field, although several recent and highly relevant references, as detailed below, are omitted, in my view. It also introduces an interesting concept, delineating the ZGA regulators into licensors and specifiers. I find this an extremely interesting and valuable opinion. However, I was expecting to find more discussion of the potential role of chromatin regulators as potential licensors, especially considering the focus of the review. For example, interference with H3K4me3 remodelling by KDM5 family demethylases leads to severe defects in ZGA (Dahl et al, PMID: 27626377, Bu et al., PMID: 35868641, Dang et al., doi: <https://doi.org/10.1101/2021.11.22.469629>), while CBP/p300 acetyltransferase activity in zygotes is critical for ZGA (Wang et al PMID: 36215692, Chan et al., PMID: 31211993) and one could potentially describe such activities as largely non-specific 'licensors'. Lastly, one section that I found lacking, considering the recent progress in this area and topic of the review, is a discussion of the nucleosome profiling/chromatin accessibility in embryos, based on recent ULI-MNase-seq, li-DNase-seq, scCOOL-seq and ATAC-seq techniques.

References missing:

Liu et al., PMID: 27626379
Du et al., PMID: 31837995
Hanna et al., PMID: 29323282
Mei et al., PMID: 33821003
Zhu et al., PMID: 36654208
Kubinyecz et al <https://doi.org/10.1101/2023.12.05.570276>
Collombet et al., PMID: 32238933
Flyamer et al., PMID: 28355183
Borsos et al., PMID: 31118510
Pal et al., PMID: 37914351
Abe et al., PMID: 36577375
Sakamoto et al., PMID: 38619966
Yang et al., PMID: 38381606

Page 17, line 481, Zscan4 is not a minor ZGA gene.

Page 11, line 328 I think it is incorrect to describe ZGA as lacking specificity. The ZGA programme involves a highly specific subset of genes.

AUTHORS' RESPONSES TO REVIEW

We deeply appreciate the Referees for their valuable comments and suggestions, which have led to significant improvement of the manuscript. We address below the issues raised by the Referees. Please note that we used Fig. 1, 2, 3, etc. to refer to figures in the manuscript and Fig. R1, R2, R3, etc. to refer to figures in this letter.

For the Referees' convenience, we also included a version of the manuscript in which the revised sections related to our responses to the Referees' comments are highlighted in blue.

Referee #1:

This is a largely comprehensive, well-written review on the interesting topic of zygotic genome activation in embryos. The authors provide the reader with a broad view of the field and information on the epigenetic changes, ZGA models and transcription factors. The greatest strength of the review is its breadth. The main weakness is that the review includes inaccuracies of the literature and omissions of relevant work. These need to be corrected before the review can be considered appropriate for publication.

Response: We thank the reviewer for the valuable suggestions. We have added the missing literature of relevant work as the Referee suggested.

In addition, the authors classify ZGA regulators into the categories of licensors and specifiers. While conceptual groupings can be helpful to simplify complex processes, the proposed use of "licensors" in this context falls short of improving the understanding of ZGA for several reasons. First, licensors apparently can be anything - components of transcriptional machinery, epigenetic regulators, metabolic regulators, nuclear pore complex - except transcription factors, which are classified as "specifiers". This means that molecules involved in any molecular events that precede transcription factor binding and RNA pol II activity are licensors - this must necessarily include nuclear assembly, chromatin assembly and even the ribosomes to facilitate translation. Therefore, the term "licensor" has no molecular meaning. Secondly, licensing as a concept has been used previously to understand the once-per-cell-cycle regulation of DNA replication. DNA replication licensing can occur only under conditions of low cyclin-dependent kinase activity and the licensing pathway was later identified as the assembly of the pre-replication complex. I would therefore argue that once the identities of proteins involved are known, the term of "licensing" becomes obsolete and one should refer to the actual protein complexes and mechanisms. Therefore, I do not think that introducing "specifiers" and "licensors" at the current time are helpful for a better understanding of ZGA.

Response: We thank the Referee for pointing out the ambiguity of these terms. These two terms were proposed to distinguish two distinct classes of factors that contribute to ZGA. We intended to use "licensor" to refer to a class of factors, rather than a specific factor. Current evidence in the field (as summarized in the review) suggests that there are multiple redundant pathways that control the timing of ZGA. We believe that it is unlikely that a single factor is responsible

for such licensing. The introduction of these terms was meant to help the discussion of ZGA regulatory factors and emphasize non-TF factors that are critical for the timing of ZGA. Nevertheless, the Referee's point is well taken and we fully agree that the current definition of licenser can cause confusion. Therefore, we introduce four criteria to better define the "licensors" for ZGA as follows (also included in the revised manuscript, lines 386-393):

- 1) A licenser is essential to initiate ZGA; depleting this factor causes severe defects in ZGA.
- 2) A licenser exhibits a unidirectional change of abundance, maturity, or localization etc. prior to ZGA and such changes generate a transcriptionally permissive state that underlies global transcription during ZGA. Preventing such changes impairs ZGA.
- 3) In principle, accelerating such changes should advance and promote ZGA. This effect however is also determined by the readiness of other licensing factors.
- 4) A licenser regulates global transcription without directly exerting impacts on the selectivity of gene activation during ZGA.

Under these new criteria, general transcription factors, "RNA polymerase II" and "epigenetic regulators" will not be classified as "licensors for ZGA" unless they exhibit gradual changes of abundance, maturation, or localization that permit ZGA to occur.

This review overlaps to some extent with Zhang & Wei, *Curr Opin Genet Dev*, "Building the genome architecture during the maternal to zygotic transition" and Du et al. Wei, *Cold Spring Harb Perspect Biol* 2022, "Epigenetic Reprogramming in Early Animal Development".

Response: The Referee is correct. The contents of this review partially overlap with the topics of the two reviews (Du et al., 2022; Zhang and Xie, 2022). Nevertheless, these two previous reviews focused on epigenetic reprogramming throughout early development, whereas we limited the focus of this review to events around and recent progress in ZGA.

Major points:

1. The schematic of 3D chromatin (Fig 2) is inaccurate for both maternal and paternal genomes and requires extensive revision. 3D genome organization comprises loops, TADs and compartments. Given that loops/TADs and compartments are antagonistic in many cell types, it is unclear how these different aspects of 3D chromatin can be summarized as one category.

Regarding maternal chromatin: 1) MII oocytes lack loops and TADs since these are condensed chromosomes, as had been shown previously for mitotic cells (Naumova et al., *Science* 2013). 2) Loops and TADs are present in maternal and paternal chromatin of one-cell embryos (Flyamer et al., *Nature* 2017) and both are dependent on cohesin in one-cell embryos (Gassler et al., *EMBO J* 2017).

Regarding paternal chromatin: Although several publications including by the authors of this review have reported the presence of TADs and compartments in mouse sperm, the work by Yin et al., *Genome Research* 2023 has convincingly demonstrated that the source of these 3D

genome features is cell-free DNA in sperm preparations. The authors point out that TADs and compartments in sperm are under debate in the figure legend, but actually the issue has been settled and this should be accurately reflected in the schematic and also openly discussed in the main text.

Response: These are excellent points. We agree that the “3D chromatin” term is too broad in this context. Therefore, we now chose “TADs and loops” as a representative feature of 3D chromatin, and changed the color bar to show that “TADs and loops” are weak in 1-cell embryos. Please see the revised Figure R1 below.

As for paternal chromatin, indeed, Yin et al. reported that compartments and TADs were absent after removal of cell-free chromatin from mature murine sperm (Yin et al., 2023). They also measured H3K4me3 and H3K27me3 in sperm, which manifested as broad domains unlike those in other cell types. However, Yin et al. also stated that “CUT&RUN, CUT&Tag, sonicated/fixed ChIP-seq, and native MNase ChIP-seq show distinct behavior following DNase or DTT treatment; together, these findings raise substantial concerns regarding affinity-based chromatin analysis in bona fide sperm and motivate continued optimization to allow more efficient recovery of proteins of interest.” These statements suggested that the authors obtained different results with different protocols even in the same lab. We have also communicated with the first author of Yin et al. and confirmed the states of histone marks remain uncertain at this moment. Therefore, although this study raised a timely alert for the community to revisit sperm chromatin data, we believe that these data still await further validation from independent labs and the true nature of sperm chromatin warrants further investigation. Therefore, we revised sperm chromatin states with “?” indicating that the topic is “under debate”. Please see Figure R1 below.

Figure R1. Epigenetic reprogramming around mouse ZGA. In mouse, ZGA is executed by RNA Pol II and facilitated by transcription factors (TFs) and chromatin regulators (CRs), amid dramatic epigenetic reprogramming. After fertilization, genome-wide loss of DNA methylation starts from the zygote stage and reaches the lowest level in blastocysts. On the paternal genome, H3K36me3, H3K4me3, H3K27me3, H3K27ac, and H2AK119ub are quickly reset after fertilization based on the globally distinct patterns between sperm and late 1-cell stage, consistent with the protamine-histone exchange that occurs shortly following fertilization. On the maternal genome, H3K36me3 is briefly inherited to 1-cell embryos, and is then reset after ZGA by marking newly transcribed genes. H3K27ac is absent from chromatin of MII eggs and is re-established around pronuclear 3-4 (PN3-4) after fertilization. H3K4me3 is briefly inherited from oocytes to early 2-cell embryos. H3K4me3 and H3K27ac then transit from non-canonical, broad domains to canonical sharp peaks during ZGA. H3K27me3 and H2AK119ub exhibit broad, non-canonical patterns in oocytes and preimplantation embryos, with H3K27me3 being inherited to blastocysts and H2AK119ub1 exhibiting more dynamic changes. Both marks switch to canonical patterns with sharp peaks at promoters after implantation (not shown). The global patterns of H3K9me3 on both alleles show distinct distributions in 1-cell embryos, indicating epigenetic resetting on both genomes. H3K9me3 at early stages is considered non-repressive and is thus marked as non-canonical. The 3D chromatin structures including loops and topologically associating domains (TADs) are substantially reduced in the MII eggs, 1-cell and 2-cell embryos but are gradually re-established after ZGA. The presence of loops, TADs, as well as the patterning of histone modifications in mouse sperm is currently under debate (Yin et al., 2023).

2. An interesting aspect of early genome architecture is that it appears to be more relaxed and the authors discuss whether this could be related to expression of CTCF (lines 305-308). It is

stated that the expression of CTCF during development cannot explain the genome architecture change during ZGA of the fly. However, this does not make sense anyhow as CTCF is known not to be present at fly domain boundaries. Therefore, fly should not be discussed in this context. Instead, it should be pointed out that CTCF is insulating TAD boundaries in 1-cell embryos as has been demonstrated by Hi-C of oocyte-specific genetic knockdown of CTCF (Dequeker et al., Nature 2022). Therefore, CTCF is constraining loop extrusion in early embryos.

Response: We thank the Referee for pointing this out. We have now removed the discussion of fly data and revised our manuscript accordingly as follows (lines 334-339 in the revised manuscript): “In human embryos, expression of CTCF, which is required for TAD formation (Chen et al., 2019), is expressed upon ZGA, correlating with the emergence of TADs. This mechanism, however, does not appear to apply to the gradual reestablishment of TADs in mouse (Du et al., 2017; Hug et al., 2017; Ke et al., 2017), as CTCF is expressed throughout mouse early development, and binds and insulates chromatin in mouse zygotes (Dequeker et al., 2022; Wang et al., 2024)”.

3. The authors describe a recent publication (Andreu et al., 2022) that uses a conditional genetic knockout approach to delete CTCF in oocytes and claim that CTCF-null embryos have largely unaffected transcription at the morula stage and develop to the blastocyst stage. CTCF, and especially the chromatin-bound fraction, is notoriously difficult to deplete. The paper does not provide any evidence that the conditional genetic knockout approach causes depletion of CTCF protein in oocytes or early embryos prior to the morula stage. It is therefore unclear whether the lack of phenotype could be due to the lack of efficient protein depletion. Indeed, the results of this paper contradict earlier work by Marisa Bartolomei. Using a genetically encoded oocyte-specific dsRNAi targeting CTCF, it was shown that CTCF loss affects gene expression in oocytes, causes defects in zygotic gene expression and an embryonic arrest around the 2- to 4-cell stage (Wan et al., Development 2008). It would be more appropriate to discuss this work in the paragraph dedicated to how genome architecture and ZGA might be related.

Response: We thank the Referee for this valuable suggestion. We revised our manuscript as follows (lines 340-346 in the revised manuscript): “The functions of CTCF and cohesin in ZGA and transcription regulation in early embryos remain a topic under investigation. Depletion of CTCF using a zona pellucida 3 (*Zp3*) promoter-driven double-stranded RNA (dsRNA) leads to defects in meiosis, development delay at 2~4 cell stage and disruption of ZGA (Wan et al., 2008). However, a more recent study reported that maternal-zygotic *Ctcf* knockout embryos can develop to the late blastocyst stage and transcription appears to be largely unaffected by the morula stage (Andreu et al., 2022). This discrepancy likely lies in the different methods of CTCF depletion.”

4. It is surprising that the authors repeatedly fail to reference the first published mouse embryo Hi-C paper (Flyamer et al., Nature 2017), when describing the 3D genome in embryos (lines 300, 302). The review should be scholarly and reference this work in both instances.

Response: We apologize for not citing this important paper. The reason was that our discussion focused on the weakened chromatin organization in zygotes compared to later stages, a topic not discussed in Flyamer et al. which focused on the chromatin organization during the oocyte-to-zygote transition. Nevertheless, we fully agree that this is a seminal publication that investigated chromatin organization in mouse oocytes and zygotes using single-nucleus Hi-C, and we have added this paper to our manuscript (lines 329-332).

5. The authors describe the data linking DNA methylation to ZGA. They state that "gametic methylation... is not expected to have a major impact on ZGA, as Dnmt3a or Dnmt3l maternal knockout (mKO) embryos, which lose all DNA methylation in oocytes, can develop beyond the pre-implantation period" (lines 96-99), yet the paragraph concludes that "proper DNA methylation reprogramming is critical for the fidelity of ZGA" (lines 113-114). These statements contradict each other. The paragraph ending needs to be toned down to reflect that DNA methylation reprogramming can promote the fidelity of ZGA.

Response: We thank the Referee for pointing this out. We intended to mean that loss of DNA methylation has a limited impact on ZGA given DNA demethylation during this period but ectopic DNA methylation could still impede ZGA, as observed in the *Stella* mutant. We have now toned down these statements and revised the concluding statement as "Taken together, these data indicate that proper DNA methylation reprogramming promotes the fidelity of gene expression during ZGA and afterward (lines 116-118 in our revised manuscript)."

6. The claim is made that "histone acetylation and deacetylation are both essential for mouse ZGA and preimplantation development" (lines 142-143). The perturbations in embryos relied on expression of dominant negatives or inhibitors and therefore indirect effects cannot be excluded. In cultured cells, the genetic engineering of 28 alleles of H3 to replace lysine 27 with arginine showed that H3K27 acetylation is not essential for gene activation, at least in the transition to epiblast-like cells (Sankar et al., Nat Genet 2022). The embryo data should be discussed in light of these findings and the conclusion quoted above needs to be toned down.

Response: We thank the Referee for pointing this out. We have now revised the statement as follows (lines 151-153 in our current manuscript): "These data suggest that histone deacetylation also contributes to mouse ZGA and preimplantation development (Fig. 2) (Wang et al., 2022b), although indirect effects from inhibitors and dominant mutants cannot be excluded."

As for Sankar et al., Nat Genet 2022 (Sankar et al., 2022), we agree that this study beautifully showed that H3K27ac alone is not essential for gene activation in this context. However, histone acetylation also occurs at a large number of lysine sites on H2A, H2B, H3, and H4, and acetylation at different sites is known to often function in a redundant manner (Kurdistani et al., 2004). Therefore, it is still possible that histone acetylation jointly regulates transcription. We have included these points in the paper (lines 139-143 in our current manuscript), "Of note, loss of histone acetylation at H3K27 was insufficient to affect gene activation in mouse embryonic stem cells (mESCs), presumably due to the presence of other histone acetylation (such as H3K9ac, H3K18ac, H4K5ac, H4K8ac, etc.) (Sankar et al., 2022;

Zhang et al., 2020). In fact, histone acetylation occurs at a large number of lysine sites and is known to exert functions often in a partially redundant manner (Durrin et al., 1991; Zhang et al., 1998)".

7. The authors should point out in the main text that the "knockout of maternally-expressed Obox1/2/5/7 and embryonically-expressed Obox3/4" (lines 520-521) was achieved by deletion of a 1.2 Mb region containing these and other pseudogenes.

Response: We revised our manuscript accordingly.

8. Towards the middle of the review, the authors start discussing work in first person, e.g. "we showed that..." (line 520). This is principally of course fine and the authors are at liberty to do so. However, it seems odd that at this point a switch to first person occurs whereas many of the other referenced work earlier was also from this group. It would be good to be consistent throughout the review.

Response: We now use consistent word usage throughout the manuscript.

Referee #3:

The authors summarized the latest findings about zygotic genome activation (ZGA), which is of broad interest. Although the ZGA has long been known, the mechanisms controlling ZGA have remained unclear. In recent years, new factors involved in the ZGA such as Dux and Obox have been found, and therefore this review about ZGA is timely and important. In addition, the authors categorize the ZGA-regulating factors into Licensors and Specifiers, enhancing the understandability of the text. The various aspects discussed in the review are relevant to a broad audience. I have a few suggestions to further improve the manuscript.

Response: We are very pleased that the Referee found our review timely and informative. We have revised the manuscript according to the Referee's suggestions.

Comments

1.107-109 The sentence flow around "However" is confusing and should be revised to increase clarity. "Unlike Stella mutants, Nlrp14 mutants exhibited defective passive demethylation but not ectopic DNA methylation. However, embryos lacking maternal NLRP14 also displayed disrupted Ca²⁺ oscillations (Meng et al., 2023), suggesting that NLRP14 has DNA methylation-independent functions."

Response: We thank the Referee for pointing this out. We modified the sentence to "NLRP14 also has DNA methylation-independent functions as embryos lacking maternal NLRP14 displayed disrupted Ca²⁺ oscillations (Meng et al., 2023)".

2.134 Fig. 1b is not present in the text. Please correct it.

Response: We apologize for the error. It should be Fig. 2. This is corrected now.

3.150 The cited reference (Chen et al. 2014 "Single-molecule dynamics of enhanceosome assembly in embryonic stem cells.") did not match the text.

Response: We are sorry for mistakenly citing this paper and have removed this reference.

4.166-169 It has been reported that the switch from non-canonical to canonical H3K4me3 via KDM5 is important for the early development of bovine and porcine embryos (Dang et al. 2022, Bu et al. 2022), which should be included.

Response: We thank the Referee for pointing this out and have added these two papers.

5.199-200 Citations are needed for these reports.

“After implantation, H3K27me3 and H2AK119ub both adopt similar distributions and canonical patterns, co-occupying developmental gene promoters (Fig. 2). Therefore, H3K27me3 appears to be briefly lost at key Polycomb target gene promoters in preimplantation embryos, leading to a resetting of the epigenetic memory from gametes to embryos.”

Response: We thank the Referee for pointing this out. We have added related citations (lines 218-219 in our current manuscript), “After implantation, H3K27me3 and H2AK119ub both adopt similar distributions and canonical patterns, co-occupying developmental gene promoters (Fig. 2) (Chen et al., 2021a; Liu et al., 2016; Mei et al., 2021; Zheng et al., 2016; Zhu et al., 2021). Therefore, H3K27me3 appears to be briefly lost at key Polycomb target gene promoters in preimplantation embryos, leading to a resetting of the epigenetic memory from gametes to embryos.”

6. 267-268 There are no references about Gap-filling of H3 variants.

“Overexpression of a dominant-negative mutant of p150, a subunit of H3.1/H3.2 chaperone CAF-1, in embryos or knocking down p150 in mouse embryonic stem cells (mESCs) both led to increased incorporation of H3.3, consistent with a “nucleosome gap-filling” mechanisms for H3.3 (Dominique Ray-Gallet et al., Mol. Cell, 2011). Upon depletion of p150, “2-cell genes” such as *Zscan4* and *Dux* were activated, suggesting that the non-canonical H3.3 in 1-cell embryos may represent a permissive chromatin environment that permits expression of minor ZGA genes (Ishiuchi et al., 2021; Ishiuchi et al., 2015).”

Response: This is now corrected as follows (lines 280-283 in our current manuscript): “Either overexpression of a dominant-negative mutant of p150, a subunit of the H3.1/H3.2 chaperone CAF-1, in embryos or knocking down p150 in mESCs led to increased incorporation of H3.3 through a “nucleosome gap-filling” mechanism for H3.3 (Ishiuchi et al., 2021; Ray-Gallet et al., 2011). Upon depletion of p150, “2-cell genes” such as *Zscan4* and *Dux* were activated, suggesting that the non-canonical H3.3 in 1-cell embryos represents a permissive chromatin environment that permits expression of minor ZGA genes (Ishiuchi et al., 2021; Ishiuchi et al., 2015)”.

7. 274-276

As far as I know, H3.1/3.2, which are canonical in mammals, are DNA replication dependent. It is unclear why fly oocytes can live with canonical H3 without DNA replication. If possible, please add an explanation regarding this.

Response: We thank the Referee for raising this excellent question, which prompted us to investigate further in the literature.

Histone H3.3 is indispensable for mammalian development as H3.3-null mouse cannot survive beyond E6.5 (Jang et al., 2015). However, H3.3 mutant *Drosophila* can survive to adulthood and appear morphologically normal, though with reduced viability. Without H3.3, the expression of canonical H3 is increased and canonical H3 is incorporated into transcribed regions in a replication-independent manner (Sakai et al., 2009). Nevertheless, both male and female mutant adult flies are sterile (Sakai et al., 2009), suggesting H3.3 is specifically required for germline development (Sakai et al., 2009). Even in this case, defects in testis development were partially rescued when overexpressing canonical H3.2 (Sakai et al., 2009), suggesting that canonical histones can replace the function of H3.3 when highly expressed (Klein and Knoepfler, 2023). Therefore, we speculate that fly oocytes may have a large amount of canonical H3 that are sufficient to support nucleosome assembly.

We have added these speculations (lines 290-299 in our current manuscript), “It is unclear why fly uses canonical H3 in oocytes. H3.3 mutant *Drosophila* can survive to adulthood and appear morphologically normal, though with reduced viability. Without H3.3, the expression of canonical H3 is increased and canonical H3 is incorporated into transcribed regions in a replication-independent manner (Sakai et al., 2009). Nevertheless, both male and female adult flies are sterile (Sakai et al., 2009), suggesting H3.3 is specifically required for germline development (Sakai et al., 2009). Even in this case, defects in testis development were partially rescued by overexpressing canonical H3.2 (Sakai et al., 2009), suggesting that canonical histones can replace the function of H3.3 when highly expressed (Klein and Knoepfler, 2023). One possibility is that flies produce canonical H3 in large quantities to compensate for H3.3 in oocytes to support the extremely rapid DNA replication cycles during embryonic development”.

8.757-760 There are duplicate citations.

Response: The error has been corrected.

9.787-788 This paper has already been published in Elife. The information of journal should be corrected.

Response: We thank the Referee for pointing this out; the information has now been updated.

10.833-836 There are duplicate citations.

Response: We apologize for the error, which has now been corrected.

11.857-862 There are duplicate citations.

Response: We apologize for the error, which has now been corrected.

12.1093-1094 This paper has already been published in Genome Research. The information of journal should be corrected.

Response: We thank the Referee for pointing this out; this information has now been updated.

13.1103-1108 There are duplicate citations.

Response: We apologize for the error, which has now been corrected.

Advisor's comments:

This review by Zou and colleagues, provides an excellent summary of the recent progress concerning the regulation of zygotic genome activation across species, from a transcriptional and epigenetic perspective. It provides a timely update of current knowledge in a rapidly advancing field, although several recent and highly relevant references, as detailed below, are omitted, in my view. It also introduces an interesting concept, delineating the ZGA regulators into licensors and specifiers. I find this an extremely interesting and valuable opinion.

Response: We sincerely thank the Referee for appreciating the value of our review, and providing valuable suggestions for these references. We have added the missing literature as the Referee suggested.

However, I was expecting to find more discussion of the potential role of chromatin regulators as potential licensors, especially considering the focus of the review. For example, interference with H3K4me3 remodelling by KDM5 family demethylases leads to severe defects in ZGA (Dahl et al, PMID: 27626377, Bu et al., PMID: 35868641, Dang et al., doi: <https://doi.org/10.1101/2021.11.22.469629>), while CBP/p300 acetyltransferase activity in zygotes is critical for ZGA (Wang et al PMID: 36215692, Chan et al., PMID: 31211993) and one could potentially describe such activities as largely non-specific 'licensors'.

Response: We thank the Referee for these valuable suggestions. We have discussed Dahl et al, PMID: 27626377, Dang et al., PMID: 35243485, Wang et al PMID: 36215692, and Chan et al., PMID: 31211993 in our previous version and have now added Bu et al., PMID: 35868641.

Although we like the idea to consider histone modifiers as licensors of ZGA, we note that histone modifications do come with certain gene and sequence specificities that may partially overlap with the concept of specifiers. Moreover, Referee 1 suggested that the licensor's definition is vague and may include many factors that are essential for ZGA, either directly or indirectly. Therefore, we have now introduced a refined licensor definition as below:

1) A licensor is essential to initiate ZGA; depleting this factor causes severe defects in ZGA.

- 2) A licensor exhibits a unidirectional change of abundance, maturation, or localization etc. prior to ZGA and such changes allow the competence of global transcription for ZGA. Preventing such changes impairs ZGA.
- 3) In principle, accelerating such changes should advance and promote ZGA. This effect however is also determined by the readiness of other licensing factors.
- 4) A licensor regulates global transcription without directly exerting impacts on the selectivity of gene activation during ZGA.

Based on this new definition, histone modifiers would not fall into this category. This point is also now discussed in our revised manuscript (lines 394-399), “It is worth noting that although epigenetic regulators often play essential roles in ZGA, they also come with certain gene specificities as individual epigenetic regulators often modulate a subset of ZGA genes with specific sequence features. For example, H3K4me3 is preferentially enriched at CpG-rich promoters (Erkek et al., 2013). Moreover, to be considered as “licensors”, they would also meet other criteria described above. For example, their nuclear or chromatin abundance is initially limited and is a bottleneck for ZGA, and its gradual changes later on permit ZGA to occur”.

Lastly, one section that I found lacking, considering the recent progress in this area and topic of the review, is a discussion of the nucleosome profiling/chromatin accessibility in embryos, based on recent ULI-MNase-seq, li-DNase-seq, scCOOL-seq and ATAC-seq techniques.

Response: We thank the Referee for this valuable suggestion. Summary of these studies and chromatin reprogramming in the entire early development was a major topic in a recent review of ours (Du et al. 2022). In this current review, we elected to focus on regulation of ZGA. In our previous submission, we discussed how key transcription factors were identified from chromatin accessibility data in mouse and human early embryos using ATAC-seq (Wu et al., 2016; Wu et al., 2018), low-input DNase I sequencing (liDNase-seq) (Gao et al., 2018; Lu et al., 2016). ULI-MNase-seq was applied to 1-cell mouse embryos 12 h right after ICSI (Wang et al., 2022a), whereas scCOOL-seq (Guo et al., 2017; Li et al., 2018) examined the entire pre-implantation development. We now briefly mentioned these technologies in the revised paper (lines 489-497).

References missing:

Liu et al., PMID: 27626379

Du et al., PMID: 31837995

Hanna et al., PMID: 29323282

Mei et al., PMID: 33821003

Zhu et al., PMID: 36654208

Kubinyecz et al <https://doi.org/10.1101/2023.12.05.570276>

Collombet et al., PMID: 32238933

Flyamer et al., PMID: 28355183

Borsos et al., PMID: 31118510

Pal et al., PMID: 37914351

Abe et al., PMID: 36577375

Sakamoto et al., PMID: 38619966

Yang et al., PMID: 38381606

Response: We thank the Referee for pointing this out. We have cited these references and added related content in our revised manuscript (lines 160-166, 184-187, 207-208, 218-219, 307-311, 323-332, 353-370, 401-403, 587-589, 603-604, 615-618, 666-667, 677-680).

Page 17, line 481, *Zscan4* is not a minor ZGA gene.

Response: We thank the reviewer for pointing this out. Nevertheless, it is unclear to us why *Zscan4* is not a minor ZGA gene. Our RNA-seq data showed that *Zscan4* and *ZSCAN4* start to be expressed in early 2-cell (E2C) embryos in mouse (which is generally considered as minor ZGA period) and 4-cell embryos in human (Fig. R2A-B) (Xiong et al., 2022; Zou et al., 2022). The expression of *Zscan4* in minor ZGA in mouse was also reported in previous papers from the Aoki lab (Fig. R2C-D) (Sugie et al., 2020), the Gao lab (Xu et al., 2023), the Zhang lab (Chen et al., 2021b), and the Trono lab (De Iaco et al., 2020), with some data included below for the convenience of reviewers.

Figure R2

Figure R2. Existing data supporting *Zscan4* as a minor ZGA gene. (A) RNA levels (Fragments Per Kilobase of transcript per Million mapped reads, FPKM) of *Zscan4b/c/d/f* detected by mRNA-seq during mouse oocyte and early embryo development (Xiong et al., 2022). (B) FPKM of *ZSCAN4* detected by mRNA-seq during human oocyte and early embryo development (Zou et al., 2022). (C) Electrophoresis images of the *Dux* family, *Zfp352*, *Zscan4d*, and rabbit α -globin (external control) PCR products (Sugie et al., 2020). (D) The band densities in (C) were quantified using ImageJ (Sugie et al., 2020).

Page 11, line 328 I think it is incorrect to describe ZGA as lacking specificity. The ZGA

programme involves a highly specific subset of genes.

Response: We thank the Referee for pointing this out. We intended to mean that the “licensors” of ZGA, but not the ZGA program, “lack specificity”. Accordingly, we have now refined the sentence to avoid confusion (lines 378-380).

References

Andreu, M.J., Alvarez-Franco, A., Portela, M., Gimenez-Llorrente, D., Cuadrado, A., Badia-Careaga, C., Tiana, M., Losada, A., and Manzanares, M. (2022). Establishment of 3D chromatin structure after fertilization and the metabolic switch at the morula-to-blastocyst transition require CTCF. *Cell reports* *41*, 111501.

Chen, X., Ke, Y., Wu, K., Zhao, H., Sun, Y., Gao, L., Liu, Z., Zhang, J., Tao, W., Hou, Z., *et al.* (2019). Key role for CTCF in establishing chromatin structure in human embryos. *Nature* *576*, 306-310.

Chen, Z., Djekidel, M.N., and Zhang, Y. (2021a). Distinct dynamics and functions of H2AK119ub1 and H3K27me3 in mouse preimplantation embryos. *Nat Genet* *53*, 551-563.

Chen, Z., Xie, Z., and Zhang, Y. (2021b). DPPA2 and DPPA4 are dispensable for mouse zygotic genome activation and pre-implantation development. *Development* *148*.

De Iaco, A., Verp, S., Offner, S., Grun, D., and Trono, D. (2020). DUX is a non-essential synchronizer of zygotic genome activation. *Development* *147*.

Dequeker, B.J.H., Scherr, M.J., Brandao, H.B., Gassler, J., Powell, S., Gaspar, I., Flyamer, I.M., Lalic, A., Tang, W., Stocsits, R., *et al.* (2022). MCM complexes are barriers that restrict cohesin-mediated loop extrusion. *Nature* *606*, 197-203.

Du, Z., Zhang, K., and Xie, W. (2022). Epigenetic reprogramming in early animal development. *Cold Spring Harbor perspectives in biology* *14*, a039677.

Du, Z., Zheng, H., Huang, B., Ma, R., Wu, J., Zhang, X., He, J., Xiang, Y., Wang, Q., Li, Y., *et al.* (2017). Allelic reprogramming of 3D chromatin architecture during early mammalian development. *Nature* *547*, 232-235.

Durrin, L.K., Mann, R.K., Kayne, P.S., and Grunstein, M. (1991). Yeast histone H4 N-terminal sequence is required for promoter activation in vivo. *Cell* *65*, 1023-1031.

Erkek, S., Hisano, M., Liang, C.Y., Gill, M., Murr, R., Dieker, J., Schubeler, D., van der Vlag, J., Stadler, M.B., and Peters, A.H. (2013). Molecular determinants of nucleosome retention at CpG-rich sequences in mouse spermatozoa. *Nat Struct Mol Biol* *20*, 868-875.

Gao, L., Wu, K., Liu, Z., Yao, X., Yuan, S., Tao, W., Yi, L., Yu, G., Hou, Z., Fan, D., *et al.* (2018). Chromatin Accessibility Landscape in Human Early Embryos and Its Association with Evolution. *Cell* *173*, 248-259 e215.

Guo, F., Li, L., Li, J., Wu, X., Hu, B., Zhu, P., Wen, L., and Tang, F. (2017). Single-cell multi-omics sequencing of mouse early embryos and embryonic stem cells. *Cell Res* *27*, 967-988.

Hug, C.B., Grimaldi, A.G., Kruse, K., and Vaquerizas, J.M. (2017). Chromatin Architecture Emerges during Zygotic Genome Activation Independent of Transcription. *Cell* *169*, 216-228 e219.

Ishiuchi, T., Abe, S., Inoue, K., Yeung, W.K.A., Miki, Y., Ogura, A., and Sasaki, H. (2021). Reprogramming of the histone H3.3 landscape in the early mouse embryo. *Nat Struct Mol Biol* *28*, 38-49.

Ishiuchi, T., Enriquez-Gasca, R., Mizutani, E., Bošković, A., Ziegler-Birling, C., Rodriguez-Terrones, D., Wakayama, T., Vaquerizas, J.M., and Torres-Padilla, M.E. (2015). Early embryonic-like cells are induced by downregulating replication-dependent chromatin assembly. *Nature structural & molecular biology* *22*, 662-671.

Jang, C.W., Shibata, Y., Starmer, J., Yee, D., and Magnuson, T. (2015). Histone H3.3 maintains genome integrity during mammalian development. *Genes Dev* *29*, 1377-1392.

Ke, Y., Xu, Y., Chen, X., Feng, S., Liu, Z., Sun, Y., Yao, X., Li, F., Zhu, W., Gao, L., *et al.* (2017). 3D Chromatin Structures of Mature Gametes and Structural Reprogramming during Mammalian Embryogenesis. *Cell* *170*, 367-381.e320.

Klein, R.H., and Knoepfler, P.S. (2023). Knockout tales: the versatile roles of histone H3.3 in development and disease. *Epigenetics Chromatin* *16*, 38.

Kurdistani, S.K., Tavazoie, S., and Grunstein, M. (2004). Mapping global histone acetylation patterns to gene expression. *Cell* *117*, 721-733.

Li, L., Guo, F., Gao, Y., Ren, Y., Yuan, P., Yan, L., Li, R., Lian, Y., Li, J., Hu, B., *et al.* (2018). Single-cell multi-omics sequencing of human early embryos. *Nat Cell Biol* *20*, 847-858.

Liu, X., Wang, C., Liu, W., Li, J., Li, C., Kou, X., Chen, J., Zhao, Y., Gao, H., Wang, H., *et al.* (2016). Distinct features of H3K4me3 and H3K27me3 chromatin domains in pre-implantation embryos. *Nature* *537*, 558-562.

Lu, F., Liu, Y., Inoue, A., Suzuki, T., Zhao, K., and Zhang, Y. (2016). Establishing Chromatin Regulatory Landscape during Mouse Preimplantation Development. *Cell* *165*, 1375-1388.

Mei, H., Kozuka, C., Hayashi, R., Kumon, M., Koseki, H., and Inoue, A. (2021). H2AK119ub1 guides maternal inheritance and zygotic deposition of H3K27me3 in mouse embryos. *Nature Genetics* *53*, 539-550.

Meng, T.G., Guo, J.N., Zhu, L., Yin, Y., Wang, F., Han, Z.M., Lei, L., Ma, X.S., Xue, Y., Yue, W., *et al.* (2023). NLRP14 Safeguards Calcium Homeostasis via Regulating the K27 Ubiquitination of Nclx in Oocyte-to-Embryo Transition. *Adv Sci (Weinh)*, e2301940.

Ray-Gallet, D., Woolfe, A., Vassias, I., Pellentz, C., Lacoste, N., Puri, A., Schultz, D.C., Pchelintsev, N.A., Adams, P.D., Jansen, L.E., *et al.* (2011). Dynamics of histone H3 deposition in vivo reveal a nucleosome gap-filling mechanism for H3.3 to maintain chromatin integrity. *Mol Cell* *44*, 928-941.

Sakai, A., Schwartz, B.E., Goldstein, S., and Ahmad, K. (2009). Transcriptional and developmental functions of the H3.3 histone variant in *Drosophila*. *Curr Biol* *19*, 1816-1820.

Sankar, A., Mohammad, F., Sundaramurthy, A.K., Wang, H., Lerdrup, M., Tatar, T., and Helin, K. (2022).

Histone editing elucidates the functional roles of H3K27 methylation and acetylation in mammals. *Nat Genet* 54, 754-760.

Sugie, K., Funaya, S., Kawamura, M., Nakamura, T., Suzuki, M.G., and Aoki, F. (2020). Expression of Dux family genes in early preimplantation embryos. *Sci Rep* 10, 19396.

Wan, L.B., Pan, H., Hannenhalli, S., Cheng, Y., Ma, J., Fedoriw, A., Lobanenkov, V., Latham, K.E., Schultz, R.M., and Bartolomei, M.S. (2008). Maternal depletion of CTCF reveals multiple functions during oocyte and preimplantation embryo development. *Development* 135, 2729-2738.

Wang, C., Chen, C., Liu, X., Li, C., Wu, Q., Chen, X., Yang, L., Kou, X., Zhao, Y., Wang, H., *et al.* (2022a). Dynamic nucleosome organization after fertilization reveals regulatory factors for mouse zygotic genome activation. *Cell Res* 32, 801-813.

Wang, M., Chen, Z., and Zhang, Y. (2022b). CBP/p300 and HDAC activities regulate H3K27 acetylation dynamics and zygotic genome activation in mouse preimplantation embryos. *Embo j* 41, e112012.

Wang, W., Gao, R., Yang, D., Ma, M., Zang, R., Wang, X., Chen, C., Kou, X., Zhao, Y., Chen, J., *et al.* (2024). ADNP modulates SINE B2-derived CTCF-binding sites during blastocyst formation in mice. *Genes Dev* 38, 168-188.

Wu, J., Huang, B., Chen, H., Yin, Q., Liu, Y., Xiang, Y., Zhang, B., Liu, B., Wang, Q., Xia, W., *et al.* (2016). The landscape of accessible chromatin in mammalian preimplantation embryos. *Nature* 534, 652-657.

Wu, J., Xu, J., Liu, B., Yao, G., Wang, P., Lin, Z., Huang, B., Wang, X., Li, T., Shi, S., *et al.* (2018). Chromatin analysis in human early development reveals epigenetic transition during ZGA. *Nature* 557, 256-260.

Xiong, Z., Xu, K., Lin, Z., Kong, F., Wang, Q., Quan, Y., Sha, Q.Q., Li, F., Zou, Z., Liu, L., *et al.* (2022). Ultrasensitive Ribo-seq reveals translational landscapes during mammalian oocyte-to-embryo transition and pre-implantation development. *Nat Cell Biol* 24, 968-980.

Xu, R., Zhu, Q., Zhao, Y., Chen, M., Yang, L., Shen, S., Yang, G., Shi, Z., Zhang, X., Shi, Q., *et al.* (2023). Unreprogrammed H3K9me3 prevents minor zygotic genome activation and lineage commitment in SCNT embryos. *Nat Commun* 14, 4807.

Yin, Q., Yang, C.H., Strelkova, O.S., Wu, J., Sun, Y., Gopalan, S., Yang, L., Dekker, J., Fazio, T.G., Li, X.Z., *et al.* (2023). Revisiting chromatin packaging in mouse sperm. *Genome Res* 33, 2079-2093.

Zhang, T., Zhang, Z., Dong, Q., Xiong, J., and Zhu, B. (2020). Histone H3K27 acetylation is dispensable for enhancer activity in mouse embryonic stem cells. *Genome Biol* 21, 45.

Zhang, W., Bone, J.R., Edmondson, D.G., Turner, B.M., and Roth, S.Y. (1998). Essential and redundant functions of histone acetylation revealed by mutation of target lysines and loss of the Gcn5p acetyltransferase. *Embo j* 17, 3155-3167.

Zhang, Y., and Xie, W. (2022). Building the genome architecture during the maternal to zygotic transition. *Curr Opin Genet Dev* 72, 91-100.

Zheng, H., Huang, B., Zhang, B., Xiang, Y., Du, Z., Xu, Q., Li, Y., Wang, Q., Ma, J., Peng, X., *et al.*

(2016). Resetting Epigenetic Memory by Reprogramming of Histone Modifications in Mammals. *Mol Cell* 63, 1066-1079.

Zhu, Y., Yu, J., Rong, Y., Wu, Y.W., Li, Y., Zhang, L., Pan, Y., Fan, H.Y., and Shen, L. (2021). Genomewide decoupling of H2AK119ub1 and H3K27me3 in early mouse development. *Sci Bull (Beijing)* 66, 2489-2497.

Zou, Z., Zhang, C., Wang, Q., Hou, Z., Xiong, Z., Kong, F., Wang, Q., Song, J., Liu, B., Liu, B., *et al.* (2022). Translatome and transcriptome co-profiling reveals a role of TPRXs in human zygotic genome activation. *Science* 378, abo7923.

Wei Xie
Tsinghua university
Medical Science Bldg D227
Beijing 100084
China

Dear Wei,

I am pleased to inform you that your review has been accepted for publication in EMBO reports. Your manuscript will be processed for publication by EMBO Press. It will be copy edited and you will receive page proofs prior to publication.

You will soon be contacted by Springer Nature to sign your publishing license. When you login to the customer service website, please use the following token to waive the article publication charges: NZEZSJY0ODG1

Should you experience any difficulty, please email publishing@embo.org.

If you have any questions, please do not hesitate to contact the Editorial Office. Thank you very much for your contribution to EMBO Reports.
